# Contrasting the roles of regional anthropogenic aerosols from the western and eastern Hemispheres in driving the 1980–2020 Pacific multi-decadal variations

Chenrui Diao[1, 2], Yangyang Xu[1, *], Aixue Hu[3], and Zhili Wang[4]

[1]Department of Atmospheric Sciences, Texas A&M University, College Station, TX, USA
[2]Department of Atmospheric Science, Colorado State University, Fort Collins, CO, USA
[3]Climate and Global Dynamics Lab, NSF National Center for Atmospheric Research, Boulder, CO, USA
[4]State Key Laboratory of Severe Weather & Key Laboratory of Atmospheric Chemistry of CMA, Chinese Academy of Meteorological Sciences, Beijing, China

*Correspondence to*: Yangyang Xu (yangyang.xu@tamu.edu)

**Abstract.** The multi-decadal variations of the Pacific climate are extensively discussed as being influenced by external forcings such as greenhouse gases (GHGs) and anthropogenic aerosols (AA). Unlike GHGs, the potential impacts of AA could be more complex because of the heterogeneity of spatial distribution during the past few decades. Here we show, using regional aerosol forcing large ensemble simulations with CESM1, that the increasing fossil fuel-related aerosol emission over Asia (EastFF) and the reduction in aerosol emission over North America and Europe (WestFF) have remarkably different impacts in driving the Pacific circulations and SST changes since the 1980s. EastFF excites a typical El Niño-like SST pattern in the tropical Pacific and weakens the climatological Pacific Walker Circulation. WestFF induces a CP-type El Niño-like SST pattern with warming at the middle region of the equatorial Pacific, which is consistent with the 2nd leading EOF pattern of the observation. Over the North Pacific region, EastFF, located at low-to-mid latitudes, favors an IPO-like SST pattern (horseshoe-like SST pattern in the North Pacific) through a teleconnection pathway between tropical and extratropical Pacific but is overwhelmed by internal variability evolving from a positive phase to a negative IPO phase. In contrast, WestFF, located at mid-to-high latitudes, strongly affects the North Pacific via a west-to-east mid-latitude pathway and induces extensive warming. The competing effects of the heterogeneously distributed regional aerosol forcings are expected to exhibit different patterns in the near future, especially the redistribution of aerosol emissions within the domain of EastFF (i.e., from East Asia to South Asia) and changes in aerosol composition. The complex future changes in anthropogenic aerosol emissions are likely to introduce more profound impacts of aerosol forcing on the Pacific multi-decadal variations.

## 1 Introduction

Greenhouse gases (GHGs) and Anthropogenic aerosols (AA) are two major external forcings that drive the long-term global and regional climate changes. In contrast to the GHG warming effect, AA shows a net cooling effect on the global mean temperature by scattering income shortwave radiation and affecting the cloud properties ((Intergovernmental Panel on Climate Change), 2018; Lin et al., 2018; Ramanathan et al., 2001). Unlike the well-mixed GHG, the AA forcings show strong spatial heterogeneity because of the uneven distribution of emission sources and relatively short atmospheric residence time (Deser et al., 2020; Diao et al., 2021; Kang et al., 2021; Shi et al., 2022). In addition, the AA forcings have also shown a non-monotonic temporal evolution. In the 1980s, the AA emissions over North America and Europe mid-latitude (hereby collectively denoted as "Western Hemisphere") reached the peak emission level and started to decrease since then due to the strengthened pollution control in the developed nations. However, the ongoing industrialization over the South and East Asia lower latitude regions (hereby denoted as "Eastern Hemisphere") has led to a nearly monotonic increase in AA emission over the past few decades. The opposite AA emission changes between the western and eastern Hemispheres effectively reduce the magnitude of the global total AA forcing and, more importantly, present a spatial shift mode from West to East since the 1980s (Deser et al., 2020; Ming and Ramaswamy, 2011; Shao et al., 2024; Shindell et al., 2015). Because of AA forcings' complex temporal and spatial features since the 1980s, many recent efforts have been made to investigate their impacts on the global and regional climates due to the recent spatial redistribution. Shi et al. (2022) separated the climate impact of AA from GHG and internal variability based on a pattern recognition method and demonstrated that the shift mode of AA forcings has dominated the total AA effects since the 1980s. Kang et al. (2021) examined the climate responses to the zonal shift (west to east) of aerosol forcing based on idealized model simulations and argued that the shift mode contributes to a La Niña-like pattern over the equatorial Pacific. However, the meridional component of AA forcing shift (from mid-latitude in the Western Hemisphere to lower latitudes in the Eastern Hemisphere) has not been closely examined. Diao et al. (2021) developed two contrasting sets of regional aerosol large ensemble simulations driven by spatially resolved historical aerosol emissions. They demonstrated the distinct roles of East versus West AA forcing in driving the zonal mean tropospheric and mid-latitude circulations, emphasizing that both zonal and meridional shifts of the AA forcing are important. More recently, Xiang et al. (2023) did idealized simulations with South and East Asia forcings separated to probe the role of AA forcing shifts within the Eastern Hemisphere. (Wang et al., 2024) indicated that the inhomogeneous aerosol forcing dominated the recent decadal change in the summertime water vapor budget over the Tibetan Plateau.

In addition to complex effects on atmospheric circulations and surface climate, recent studies suggested that the spatially heterogeneous and temporally non-monotonic AA forcing could also modulate oceanic variations at interannual to multi-decadal time scales, which were conventionally thought to be mainly driven by internal variability. Booth et al. (2012) argued that the decline of European AA emissions contributes to the Atlantic multidecadal variations, and such arguments

have been supported by several following studies (Bellomo et al., 2018; Hassan et al., 2021; Hua et al., 2019; Watanabe and Tatebe, 2019). However, to what extent AA forcing drives the observed Atlantic decadal variations, relative to the internal variability remains highly debated. In addition, studies have focused on the AA effects on the Pacific variations (Allen et al., 2014; Dittus et al., 2021; Dong et al., 2014; Hua et al., 2018; Meehl et al., 2021), showing that AA forcings can induce decadal variations over both tropical and extratropical Pacific. Shi et al. (2023) investigated the impact of AA forcing on the North Pacific by looking at the subsurface ocean temperature responses and found unique zonal-mean patterns, which further indicated the importance of AA forcing in the North Pacific climate. However, the mechanisms of how regional AA forcings affect the extratropical Pacific remain less investigated due to the entangled offsetting effects from East and West AA.

Based on the large ensemble of regional aerosol simulations conducted by Diao et al. (2021), here we aim to isolate and contrast the climate responses due to the regional aerosol forcings originating from the eastern to western Hemispheres. We untangle their offsetting effects over the tropical and extratropical Pacific (especially North Pacific), and further investigate the dynamical mechanism of how regional AA forcings remotely affect the extratropical North Pacific regions in a multi-decadal time scale.

## 2 Methods

### 2.1 Model and numerical experiments

The model simulations in this study are based on the Community Earth System Model 1 (CESM1) fully coupled model developed by the National Center for Atmospheric Research (NCAR) and community scientists (Hurrell et al., 2013). We leverage the CESM1 large ensemble (Kay et al., 2015) and CESM1 single forcing large ensembles (Deser et al., 2020) to investigate the overall climate effects of the combined external forcings (abbreviated as "ALL"), industrial aerosols (fossil-fuel-source related, abbreviated as "FF"), and biomass-burning related aerosols (abbreviated as "BMB").

In addition, considering the opposite aerosol emission trends between the eastern and western Hemispheres, two sets of large ensemble regional AA forcing simulations (Diao et al., 2021) were applied, which follows the setup of CESM1 single forcing large ensemble. The regional simulations separated climate impacts induced by the increasing FF over East and South Asia ("Fix_EastFF1920" in Diao et al. (2021)) and by the decreasing FF over North America and Europe ("Fix_WestFF1920" in Diao et al. (2021)). Both of the regional AA forcing simulations cover 1980 to 2020 with 10 ensemble members each. The simulations follow the design of the CESM1 single forcing large ensemble; that is, fixing the AA forcing over focused regions at 1920 levels while keeping outside AA and all other external forcings evolving with time into the 21st century. Thus, the climate response to the regional aerosol forcings can be isolated by subtracting the regional simulation results from "ALL" results. The difference between ALL and the two regional AA simulations are denoted as "EastFF" and "WestFF" hereafter, respectively. As an example, Figure 1a–c shows the decadal changes in aerosol optical

depth (AOD) induced by EastFF, WestFF, and FF from 1980 to 2020, respectively. One note here is that the climate changes in response to FF do not necessarily equal the simple sum of that in response to EastFF and WestFF (denoted as EastFF+WestFF) because FF can contain potential nonlinear interactions between EastFF and WestFF impacts. Additionally, the FF results also contain aerosol forcings originating from other regions not covered by EastFF and WestFF (e.g., Africa and Arabian Peninsula, Fig. 1d), even though their magnitude is considerably smaller compared to the aerosol forcings in EastFF and WestFF. More details of the regional AA single forcing large ensemble simulations are described in Diao et al. (2021).

All analyses in this study are based on the ensemble-averaged results of the monthly outputs of the five experiments mentioned above (ALL, FF, EastFF, WestFF, and BMB) to exclude the impact of randomly generated internal variability in the model. Annual means are calculated prior to analyses.

## 2.2 Observational data

In order to test the fingerprints of regional aerosol forcings in real-world observations, the ECMWF Reanalysis v5 (ERA5) monthly reanalysis dataset developed by the European Centre for Medium-Range Weather Forecasts (ECMWF) is applied in this study (Hersbach et al., 2023). ERA5 data provides global monthly sea surface temperature (SST) anomalies from 1940 to the present day at a 0.25° x 0.25° grid. We utilized the period of 1980 to 2020 from ERA5 to be consistent with the model simulation coverage. To facilitate comparisons between the CESM simulations and the reanalysis results, we interpolate the ERA5 data onto the model grid of 1° latitude x 1° latitude. We also tested another widely used SST reconstructed dataset – the Extended Reconstructed Sea Surface Temperature (ERSST; Huang et al., 2017) – and got consistent decadal variation patterns over the Pacific, so we only show the ERA5 results in this study.

## 2.3 Metrics for Pacific Decadal Variations

To investigate the potential impact of regional aerosols on the Pacific decadal variations (PDV), we use the conventional definition of the Interdecadal Pacific Oscillation (IPO) index as the leading empirical orthogonal function (EOF) of the SST anomalies over the Pacific Ocean (60° S–70° N, 110° E–70° W) during 1980–2020 (Meehl et al., 2009). An 11-year low-pass filter is applied to the SST anomalies prior to the EOF analyses in order to obtain the interdecadal variability. For the model simulation results, we subtract the global averaged SST time series from the simulated SST patterns prior to the EOF analysis to remove the global warming mode induced by the greenhouse gases (GHG). For ERA5, the detrended SST is obtained by subtracting the least squares linear trend of the long-term SST (Xu and Hu, 2018). In addition to the EOF-based IPO index, we also use the Triple index for the Interdecadal Pacific Oscillation (TPI) developed by (Henley et al., 2015) as the time evolution of the PDV variation. TPI is defined as the difference in SST anomalies between the central Equatorial Pacific (10° S–10° N, 170° E–90° W) and the average of the western North Pacific (25°–45° N, 140° E–145° W) and western South Pacific (50°–15° S, 150° E–160° W). A positive (negative) TPI index indicates that the IPO is a positive

(negative) phase. A 13-year Lanczos low-pass filter is applied to the SST anomalies prior to the calculation of IPO and TPI index to smooth out the high-frequency signals such as ENSO.

In addition to the IPO pattern, the 2nd leading EOF of the observed Pacific SST anomalies shows an inter-hemispheric asymmetry pattern over the eastern tropical Pacific (ETP). To examine whether the regional AA forcings have any impacts on this asymmetric pattern, we define an ETP Asymmetric index, which is similar to the definition of TPI but calculates the difference in smoothed SST anomalies between the north-eastern Tropical Pacific (0°–20° N, 120° W–100° W) and south-eastern Tropical Pacific (0°–20° S, 120° W–100° W).

## 3. Results

### 3.1 Pacific SST responses to AA forcings

Since the 20th century, industrial aerosols generated from fossil fuel sources ("FF"), dominated by sulfate aerosols, have induced an overall net cooling effect globally. However, the emission control in the Western Hemisphere has led to a continuous weakening of the aerosol cooling effects, which effectively induced large-scale sea surface warming over the
Northern Hemisphere (NH) since the 1980s. Looking at the spatial pattern of the SST changes (Fig. 2c), significant warming mainly occurs at the NH mid-to-high latitudes, which is consistent with the latitudinal ranges of the emission cut region in the Western Hemisphere. Interestingly, over the extratropical North Pacific, where no local FF aerosol is emitted or removed, there is still a significant warming trend, similar to the North Atlantic (Fig. 2c). This reveals that the FF aerosols not only impact the local climate change over the emission domain but also impose a strong remote influence over the North
Pacific. However, based on previous studies (Diao et al., 2021; Kang et al., 2021), the opposite aerosol forcing evolutions from east (continuously cooling) and west (net warming trend due to emission cuts since the 1980s) may play competing roles over the North Pacific, which is what we will separate using the regional forcing simulations mentioned above.

Based on results from the two regional AA experiments (Figs. 2a & b), it is clear that the Pacific SST responds differently to
EastFF and WestFF. Over the North Pacific, the EastFF favors a PDO-like warming pattern at a multi-decadal time scale, with tropical Eastern Pacific warming (Fig. 2a). The horseshoe-like SST pattern occurs over the North Pacific, with warming over the eastern part of the North Pacific basin and high latitude while cooling over the western basin of the North Pacific. This is expected due to the advection of aerosols (especially from East Asia) from land to the North Pacific (Fig. 1a) and may also be related to the teleconnection between tropical and extratropical Pacific (Gong et al., 2006; Wilcox et al., 2019)
The strong horseshoe-like SST pattern in the North Pacific, coupled with counter-clockwise wind anomalies, intensifies the Aleutian Low.

Smith et al. (2016) argued that the AA forcing originating from the East Asia region induces large-scale warming in the North Pacific and leads to a weakening of the Aleutian Low, while our results here, however, do not support such an argument. The reason is that only the total aerosol-only experiments from CMIP5 were analyzed in the previous study, which indeed yields consistent responses when compared to the FF+BMB results in the CESM1 single forcing large ensemble experiments (Deser et al.(2020); also shown in Fig. S1 in the supplement despite a weaker magnitude). Therefore, we suggest the nonlinearity when combining regional (e.g., EH and WH) or sectoral (e.g., FF and BMB) aerosol responses makes it very challenging to clearly distinguish the climate impact due to aerosol forcings from particular sources or regions simply based on those total "aerosol-only" experiments as in CMIP5/6's DAMIPs, hence justifying the importance of running more nuanced regional aerosol perturbation experiments as designed and conducted here in our study.

Based on the regional AA forcing experiments, it is clear that the large-scale warming in the North Pacific is not only influenced by the EastFF but also strongly driven by the WestFF (Fig. 2b), which will be discussed in detail later. The extratropical South Pacific also shows a similar pattern as the North Pacific in response to EastFF but with a smaller magnitude. In the Tropical Pacific, EastFF induces a typical El Niño-like SST pattern, with cooling over the Indo-Pacific warm pool and the tropical Indian Ocean while warming over the Central and Eastern Equatorial Pacific. This is consistent with the results of Verma et al. (2019), except that they focused on the SST response induced by volcanic aerosol emissions from Asia. In response to the zonal SST gradient over the tropics, the climatological easterly wind over the equatorial Pacific is significantly weakened by westerly anomalies induced by EastFF (Fig. 2a). Similarly, the equatorial westerlies over the Indian Ocean are also weakened but by a smaller magnitude, effectively weakening the climatological Walker circulation. The strongest SST cooling induced by EastFF occurs over the North Pacific mid-latitudes far away from the emission domain, which again highlights the importance of aerosol remote forcing over the North Pacific.

In contrast to EastFF, WestFF induces large-scale warming over the entire extratropical North Pacific (Fig. 2b), which completely offsets the EastFF-induced cooling and dominates the warming trend captured in the FF case (Fig. 2c). Compared to the North Atlantic warming (the subject of many previous studies, e.g., (Booth et al., 2012; Fiedler and Putrasahan, 2021), the North Pacific warming induced by WestFF is surprisingly even stronger, indicating the significant teleconnection of WestFF remote forcing onto the North Pacific. The strong warming due to WestFF also induces local clockwise wind anomalies over the Aleutian low, which also dominates the FF-induced response (Fig. 2c). The WestFF-induced warming pattern in the North Pacific is similar to some internal variability experiments from other extratropical teleconnection experiments in previous studies, which indicated the extratropical teleconnection between the Atlantic and the Pacific (Meehl et al., 2021; Ruprich-Robert et al., 2017).

The weakening of aerosol forcing in the Western Hemisphere is widely recognized to contribute to North Atlantic warming or even affect the AMV (Dong and Sutton, 2021; Fiedler and Putrasahan, 2021; Watanabe and Tatebe, 2019). However, the

tropical teleconnection processes emphasized in the previous studies, where North Atlantic warming induces an El Niño-like SST pattern over the Tropical Pacific, are completely absent in our simulation results (Fig. 2b), presumably due to the lack of warming over the equatorial Atlantic Ocean that could excite the tropical teleconnection bridge (Wang, 2019). This reveals that an accurate and realistic characterization of the latitudinal placements of aerosol forcings determines how it remotely contributes to the North Pacific climate responses.

In the Tropical Pacific, Equatorial SST pattern in response to WestFF shows the greatest warming over the central basin of the Equatorial Pacific and cooling over south-eastern basin (Fig. 2b), similar to a central Pacific (CP) type El Niño pattern (different from the typical El Niño-like pattern induced by EastFF in Fig. 2a), which is found to be more frequently occurring in recent decades (Freund et al., 2019). In addition, WestFF also induces inter-hemispheric asymmetry of SST over the eastern tropical Pacific, with significant warming in the north-eastern tropical Pacific and slight cooling over the cold tongue region (Fig. 2b), which again is the dominant feature of FF simulation and would be completely missed in the simple linear combination of EastFF and WestFF response (Fig. 2d).

The linear summation of WestFF and EastFF results (denoted as "WestFF+EastFF") presented in Fig. 2d shows greater SST responses in the tropical Pacific and tropical Indian Ocean compared with actual FF results (Fig. 2c). However, most of the warming signals over the central and eastern tropical Pacific calculated from WestFF+EastFF are statistically insignificant. Notably, the equatorial West Pacific (160ºE–180º) exhibits a significant warming signal in WestFF+EastFF, which is missing in FF results. This disagreement between FF and WestFF+EastFF is likely due to the nonlinear interactions between the impacts of EastFF and WestFF. Similarly, in the extratropical North Pacific, WestFF+EastFF closely resembled the EastFF pattern (i.e., the cooling trend in the western extratropical Pacific; Fig. 2a) while the actual FF results are dominated by WestFF (Fig. 2b&c). The Atlantic response appears to be largely consistent despite greater magnitudes in WestFF+EastFF, with warming from both EastFF and WestFF. Besides the nonlinearity issue, aerosol forcings outside the two focused regions (i.e., aerosols in Africa and the Arabian Peninsula; see Fig. 1d) could also partially contribute to the differences between FF and WestFF+EastFF, particularly by driving cooling over the western tropical Indian Ocean and weakening the wind anomalies (Fig. 2c&d). Additionally, the aerosols outside the focused region could also impose a remote impact on the tropical Pacific region (Huang et al., 2021; Shi et al., 2022), but such impacts are likely to be smaller compared to the nonlinear interactions between EastFF and WestFF impacts, given the small magnitude of the radiative forcing (Fig. 1d).

### 3.2 Possible AA impacts in observed Pacific variations

As discussed in Sect. 3.1, EastFF induces an IPO-like SST pattern over the Pacific featuring a positive IPO phase (warming tropical eastern Pacific), while WestFF induces large-scale warming in the extratropical North Pacific and favors CP-type El

Niño SST pattern with inter-hemispheric asymmetry in the tropics. Thus, it is important to ask whether and how the regional aerosol forcings might have affected the observed Pacific multi-decadal variations from 1980 to 2020.

Figure 3a–b presents the two leading EOF patterns of the observed SST anomalies from ERA5, in which the leading EOF pattern (Fig. 3a) and the corresponding principal component are defined as the IPO pattern and IPO index, respectively. We note that both the EastFF-induced Pacific SST pattern (Fig. 2a) and the leading EOF pattern of EastFF SST responses (Fig. 3c) resemble the conventional IPO pattern (with an uncentered pattern correlation of 0.43), suggesting that the traditional calculation of IPO can be partly affected by the EastFF. The increasing TPI index induced by EastFF indicates that EastFF favors a positive phase of IPO (blue curve in Fig. 3e; the traditional EOF-defined IPO index resembles the TPI index and thus is not shown). However, the observed IPO shows a transition from the positive to negative phase from 1980 to 2020 (black curve in Fig. 3e), which is opposite to that induced by EastFF. This suggests that EastFF weakened the negative evolution of the observed IPO in terms of magnitude during this period, but the contribution is small and is overwhelmed by the observed IPO. One thing to be noted here is that the observed IPO is primarily considered to be dominated by internal variability, although other external forcings (such as volcanic aerosols) might also correspond to the observed transition. Furthermore, the modulation of external forcings to the internal variation is a complicated question, which can also involve the impacts of frequencies, magnitude, and trends. In this study, however, we are primarily focused on the multi-decadal trend of the Pacific SSTs driven by the regional aerosol forcings. Further detailed analyses on the interaction between external forcings (including, but not limited to, aerosol forcings) and internal variability are worth more investigation, but are beyond the scope of this study, though.

In contrast to the EastFF, WestFF favors an inter-hemispheric asymmetric pattern over ETP, with warming in the NH part while cooling in the SH part (Fig. 2b), which is similar to the second EOF pattern (Fig. 3b; with a pattern correlation of 0.53). Comparing the ETP asymmetric index obtained from ERA5 and WestFF, the WestFF-induced asymmetric response shows the same evolution tendency as the observed variation (Fig. 3f). It is clear that both the WestFF-induced pattern and the second EOF pattern show a CP-type El Niño-like SST pattern in the equatorial Pacific, as well as the inter-hemispheric asymmetry in ETP cold tongue region. This indicates the potential role of WestFF in driving the secondary mode of Pacific variation at decadal to multi-decadal timescales. Recent studies also demonstrate the observed asymmetric changes in ENSO activities (Jiang and Zhu, 2018), which could also be partly affected by the WestFF. However, the interannual variation is beyond the scope of this study.

One caveat to be noted is that the aerosol forcing scenario (RCP8.5) used in both the Fix_EastFF1920 experiments and ERA5 reanalysis dataset has been shown to overestimate the aerosol emission level in East Asia and miss its observed reduction since the early 2010s despite remaining at a high level (Wang et al., 2021b; Xiang et al., 2023). This leads to the overestimation of the EastFF forcing in our model experiments than the real world. On the other hand, similar aerosol

forcing biases exist in the ERA5 reanalysis dataset (Hersbach et al., 2020), although the negative impact is mitigated by actually assimilating radiation flux measure and surface temperature. Therefore, although the simulation results and ERA5 appear largely comparable, cautions should be taken in using them to quantitatively interpret the remote impact of Asian aerosol on the North Pacific in the recent couple of decades. In addition, the South Asia emission trend largely follows the assumed emission scenario, which leads to a dipole of aerosol forcings changes within the EH. The forcing dipole might introduce complex circulation responses and lead to different responses over the North Pacific. Simulations with more updated aerosol emission inventory and forcing trends and new observational datasets are necessary to fully explore the realistic climate responses to Asian aerosol forcings.

### 3.3 Walker circulation responses to AA forcings

Motivated by the distinct impacts of EastFF versus WestFF on the tropical Pacific SST, here we further examine zonal circulation responses.

Figures 4a and c show the equatorial zonal circulation and precipitation changes in response to EastFF. Induced by the local increasing aerosol emission, the maritime continent receives less surface heat flux (not shown) and weakens the Pacific zonal SST gradient from west to east, concurrent with downward motion anomalies over the maritime continent (Fig. 4a). However, over the Equatorial West Pacific (EWP), where a significant warming signal occurs, there are strong upward motion anomalies along with increases in precipitation. Together, these induce counterclockwise zonal circulation anomalies within the ascending branch of the climatological Pacific Walker circulation (PWC). Over the central and eastern equatorial Pacific (descending branch of PWC), EastFF largely induces weak upward motion anomalies, effectively leading to the weakening of PWC. Overall, although having a similar SST pattern with the El Niño events, EastFF induces a weakening of PWC but with the strongest impact *within* the ascending branch by shifting the ascending center eastward. As a minor note, EastFF also induces weak descending motion over the equatorial Atlantic Ocean by inducing trans-basin westerlies at a higher level, and thus weakens the Atlantic Walker circulation (AWC; Fig. 4a). This is consistent with the tropical teleconnection pathway between the tropical Pacific and tropical Atlantic (Meehl et al., 2021).

Despite being a remote forcing, WestFF induces very similar zonal circulation changes over the Indo-Pacific warm pool region compared to EastFF (Fig. 4b). This is consistent with the CP-type El Niño pattern (Fig. 2b) with a positive SST zonal gradient within the Indo-Pacific warm pool regions. However, the WestFF response shows opposite circulation changes in the descending branch of climatological PWC, which strengthens the downward motion and suppresses the local rainfall (Fig. 4d). Consistent with the latitudinally asymmetric SST pattern over the eastern Equatorial Pacific (Fig. 2b), the Walker Circulation and precipitation responses also feature asymmetric changes with a northward shift. Overall, the WestFF enhances the strength of the PWC and slightly shifts the ascending center of the PWC eastward. Surprisingly, the WestFF does not induce any noticeable Walker circulation changes over the equatorial Atlantic, despite the warming trend over the

tropical/subtropical Atlantic Ocean (Fig. 2b). This further indicates that the tropical Pacific responses to WestFF are not facilitated by the tropical teleconnection pathway. The zonal gradient of SST between the Indo-Pacific warm pool and the eastern basin of the Equatorial Pacific is commonly utilized to indicate the strength of PWC. However, the zonal gradient cannot fully indicate the changes in PWC induced by regional aerosol forcings because of the zonal shift of the ascending branch of PWC. This implies the importance of considering the zonal placement of radiative forcing, especially for short-lived aerosol forcings.

Similar to the SST changes, a strong nonlinear interaction between the impacts of EastFF and WestFF in the equatorial Pacific also occurs in FF results, where no significant PWC responses are exhibited in the Pacific (Fig. S2 in the supplement). Instead, FF induces a slight weakening of Walker circulations over the Indian Ocean and the Atlantic, which is consistent with the low-level wind anomalies shown in Fig. 2c.

### 3.4 Distinct teleconnection mechanisms in driving the North Pacific

As demonstrated in Sect. 3.1, the extratropical North Pacific has the largest warming in response to aerosol forcings remotely. Both EastFF and WestFF contribute to the warming in the eastern North Pacific but offset each other over the western basin of the North Pacific. This brings the question of whether and how the East and West aerosol forcings affect the North Pacific climate pattern differently. Previous studies have demonstrated the teleconnections between the Pacific and Atlantic as well as the connections between the tropical and extratropical Pacific, but most of these studies are in the context of understanding the internal variability of the Pacific or the Atlantic (Kang et al., 2021; Meehl et al., 2021; Wang, 2019; Yao et al., 2022). Here, we aim to demonstrate a unique mechanism of how regional aerosols affect the North Pacific mid-latitude climate remotely.

Figure 5 shows the upper troposphere eddy geopotential height (Z200e) and stream function changes in response to the regional FF forcings and the corresponding sea level pressure (SLP) changes. The EastFF-induced radiative cooling over the Indo-Pacific warm pool increases the local SLP, which leads to the weakening of PWC (Fig. 4a) and westerly wind anomalies at the near-surface level. As a result, the readjustment of PWC and low-level wind excites an El Niño-like SST pattern over the tropical Pacific (Fig. 2a), which excites a Rossby wave train from the tropics to extratropic (Fig. 5a). The enhanced convection over the central Equatorial Pacific (Fig. 4a) leads to anomalous upper-level high in the tropical North and South Pacific, and anomalous upper-level low over the extratropical North Pacific (Fig. 5a). Correspondingly, as a barotropic system, the SLP also shows anomalous low over the North Pacific and cyclonic near-surface wind anomalies (Fig. 5e). As another contributing factor, the emitted aerosols over East Asia (Fig. 1a) are transported to the western basin of North Pacific (Booth et al., 2012; Diao et al., 2021; Xiang et al., 2023), which induces cooling effects and further enhances the low-level westerly wind anomalies in the subtropical North Pacific. The intensified westerly wind at the mid-latitudes increases the surface heat flux and, thus, induces a cooling response over the western North Pacific, following the wind-

evaporation-SST (WES) feedback (Xie, 1996). This explains why EastFF induces the IPO-like pattern over the extratropical North Pacific region; indeed, the mechanism here in the EastFF case largely resembles the classic ENSO teleconnection originating from the tropical and propagating into extratropical Pacific, but here manifesting at a multi-decadal time scale.

Next, we show that the patterns of surface and upper-level geopotential height in response to WestFF are remarkably different from those in response to EastFF, where no obvious upper-level wave activity is found in the tropics (Fig. 5d). The radiative heating in the Western Hemisphere mid-latitude area excites wave trains propagating into the NH mid-latitudes, resulting in anomalous upper-level high over the extratropical North Pacific (Fig. 5d). As a result, SLP rises over the eastern North Pacific (Fig. 5f), with anticyclonic near-surface wind anomalies. In contrast to the EastFF response, such anomalous easterlies in the mid-latitudes surface weaken climatological westerly wind, reducing the energy advection via WES feedback (Sun et al., 2017; Wang, 2019). This can amplify the significant warming over the eastern basin of the North Pacific in response to WestFF. The dynamical responses in FF largely follow the mid-latitude pathway in WestFF (Fig. 6a, e, &e), whereas the tropical Pacific shows no significant changes in Z200e. This is consistent with the surface patterns where WestFF dominates the North Pacific warming and tropical Pacific exhibits insignificant temperature changes (Fig. 2c). In addition, the North Atlantic in FF exhibits a significant decrease in sea level pressure, which is absent in either WestFF or EastFF.

Some previous studies demonstrated that the North Atlantic warming remotely impacts the North Pacific via the tropical connections between the Atlantic and the Pacific (Johnson et al., 2020; Meehl et al., 2021; Sun et al., 2017). However, the tropical teleconnection pathway does not seem to be in play in the WestFF case given that there are no significant Atlantic Walker circulation changes (Fig. 4b) or geopotential height perturbations over the tropical Pacific (Fig. 5d). The mid-latitude teleconnection pathway invoked by WestFF aerosol forcing that we propose here is less discussed in previous studies. We argue that this west-to-east mid-latitude pathway is the dominant pathway for the perturbations introduced by regional forcing placed at higher latitudes.

Evidence also suggests that the teleconnection pathways induced by both EastFF and WestFF are predominantly associated with wintertime wave responses in the extratropics (details see Fig. S3 in the supplement). Recent studies demonstrated that the impacts of anthropogenic aerosol forcings on regional precipitation are heterogeneous and seasonal dependent (e.g., Allen and Zhao, 2022; Samset, 2022). The distinct pathways of EastFF and WestFF are likely to introduce complex regional precipitation patterns that vary seasonally, which warrants further investigation in future studies.

This sensitivity to the latitudinal displacement of forcing is supported by our secondary analysis of the biomass burning-related aerosol experiment (BMB), in which one major aerosol forcings are located in northeastern Asia in this particular model experiment (Fig. 2 in Diao et al., 2021), and we find similar wave trains propagating in the mid-latitudes (Fig. 6b;

check Fig. S4 in the supplement for wintertime wave responses). Similarly, in some other model experiments, the Atlantic
heating, when placed in the extratropical, regardless of internal variability (Yao et al., 2022) or external forcings (Ruprich-Robert et al., 2017), does not excite the tropical teleconnection pathway. This further highlights the sensitive role of the latitudinal location of forcings. The BMB forcing excites a wave train propagating in the mid-latitudes, which later impacts the lower latitudes remotely. In fact, studies show that BMB, in addition to FF, also plays an important role in driving long-term climate variations (Allen et al., 2024; Fasullo et al., 2022; Tian et al., 2023; Yamaguchi et al., 2023), and more detailed
analyses on the climate impacts of BMB on Pacific variations are warranted. However, since the focus of this study is fossil fuel-related aerosol emissions, we leave such explorations to future work.

In summary, although EastFF and WestFF both remotely drive the long-term SST changes in the North Pacific, due to the different latitudinal locations of the forcings, there are remarkably different teleconnection pathways. Therefore, the decadal
changes in the North Pacific SST could be hard to understand if only looking at the total aerosol forcing experiments.

## 4. Conclusion and discussion

Since the 1980s, the radiative forcing due to industrial-related aerosol emission has shown opposite emission trends over the Eastern and Western Hemisphere (Kang et al., 2021). Using a set of large ensemble transient simulations, we find that aerosol forcings in both the eastern and the western Hemispheres can induce multi-decadal climate responses over the
tropical and extratropical North Pacific but with distinct mechanisms.

Over the tropical Pacific, EastFF (cooling trend) induces a typical El Niño-like SST pattern and anomalous westerly wind over the tropical Pacific. Corresponding to the adjustment of the equatorial SST gradient, EastFF drives an eastward shift of the ascending branch of the PWC and an upwelling motion anomaly in the descending branch of the PWC, which effectively
weakens the overall strength of the climatological PWC. In contrast, WestFF (net warming trend since the 1980s) induces a CP-type El Niño-like SST pattern in the tropical Pacific with the most profound warming over the western and central equatorial Pacific but cooling over the eastern equatorial Pacific. At the same time, the eastern equatorial Pacific also shows inter-hemispheric asymmetry in response to WestFF, which resembles the 2nd EOF pattern of the Pacific SST changes. WestFF drives an eastward shift of the ascending branch of the PWC, which is similar to EastFF but strengthens the broader
descending branch of the PWC.

EastFF imposes a positive IPO-like pattern onto the Pacific. However, the observational IPO during 1980–2020 features a shift to the negative phase, which indicates that internal variability is the dominant driver of the observed IPO pattern. This suggests the EastFF-induced positive IPO pattern partly offsets the internal variability from 1980 to 2020. The inter-
hemispheric asymmetry pattern (stronger warming in the North Pacific) driven by WestFF resembles the 2nd leading EOF

pattern of the Pacific SST changes, suggesting that WestFF at least partly contributes to the observed asymmetric pattern over the Eastern Tropical Pacific during 1980-2020.

In the extratropical North Pacific, which is the focus of this study, EastFF leads to a positive IPO-like SST pattern, with cooling over the western part and warming over the eastern part. In contrast, WestFF induces large-scale warming in the North Pacific, which is even stronger than in the North Atlantic. This warming almost completely offsets the cooling driven by EastFF in the overall FF case. EastFF remotely drives the North Pacific following the typical ENSO teleconnection pathway between tropical and extratropical Pacific but at a multi-decadal time scale. The commonly discussed tropical bridge pathway between the Atlantic and the Pacific is not found to be important in the WestFF responses. Instead, we show that WestFF excites the North Pacific responses via a mid-latitude teleconnection pathway, which has been discussed less in previous studies. The distinction of teleconnection mechanisms is because the WestFF forcings are located at higher latitudes (Diao et al., 2021). This argument is supported by the result of the BMB aerosol forcing experiment, which is displaced even further in the higher latitude in NH (Fig. 6b, d, & f). This further suggests the importance of the latitudinal location of the aerosol forcings.

Based on the single model large ensemble method, the simulations applied in this study effectively separate the externally forced climate responses from the model-generated internal variations (Deser et al., 2020; Diao et al., 2021; Kay et al., 2015). However, one limitation of this study is that all results shown in this study are purely based on a single climate model (i.e., CESM1), which for sure includes model biases. Although CESM1 is proven to have a good performance in aerosol simulations, it has a relatively larger aerosol effective radiative forcing among climate models (−1.37 W m−2 based on Deser et al., 2020). However, the recently established Regional Aerosol Model Intercomparison Project (Wilcox et al., 2023) introduces a new multi-model framework to explore the climate impacts of regional aerosols. Further analyses similar to what is covered in this study based on the multi-model simulations in RAMIP are worth conducting to test the robustness of the conclusions presented in this study.

In this study, we focus on the period of 1980–2020, when aerosol emissions over Asia show an overall increasing trend. However, studies have shown that the emission from East Asia reached a peak around the 2010s and started to decline recently while South Asia emission continues to increase (Ramachandran et al., 2020; Samset et al., 2019; Wang et al., 2021a). The sensitivity of aerosol forcing's latitudinal placement within Asia is also highlighted by recent studies on dipole patterns emerging in Asia (Wang et al., 2022; Xiang et al., 2023). Also, the current offsetting effects between EastFF and WestFF can flip to a joint effect over specific regions of the North Pacific in the coming few decades. Such decline and/or redistribution of aerosol emissions can lead to distinct climate responses locally and remotely, which demands further continuous investigation in the future.

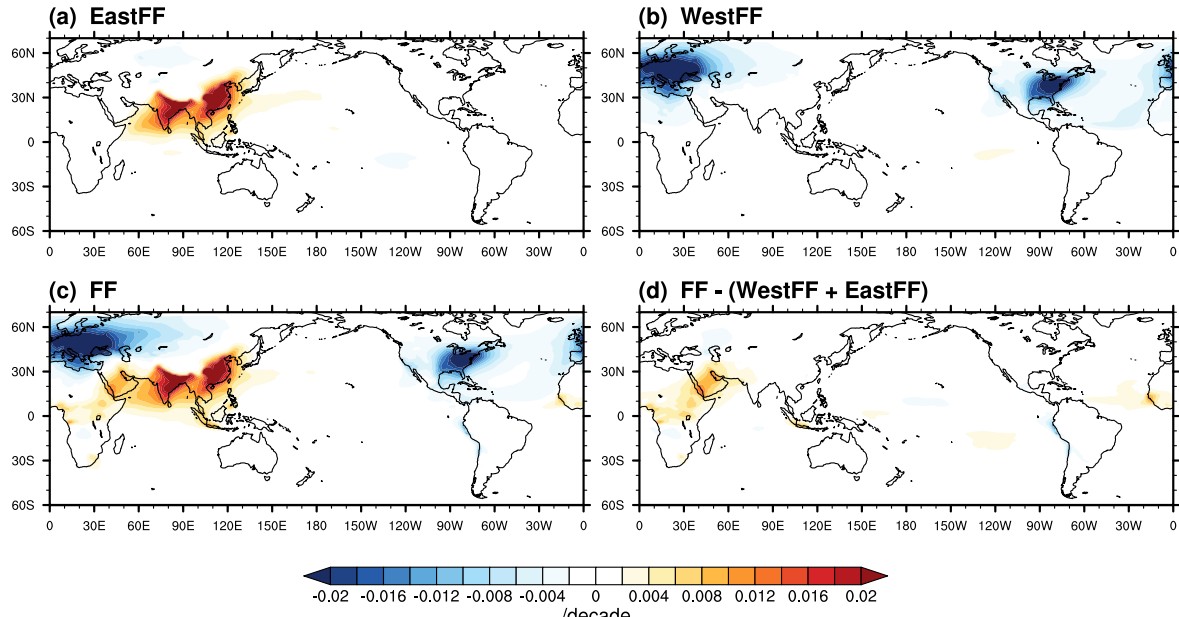

**Figure 1**

Decadal trends in anthropogenic aerosol optical depth at 550 nm (AOD-AA; 1 per decade) during 1980–2020 in response to (a) EastFF, (b) WestFF, and (c) FF. AOD-AA is calculated by subtracting the dust AOD from the total AOD. (d) The difference between FF-induced AOD changes and the combination of AOD changes induced by EastFF and WestFF.

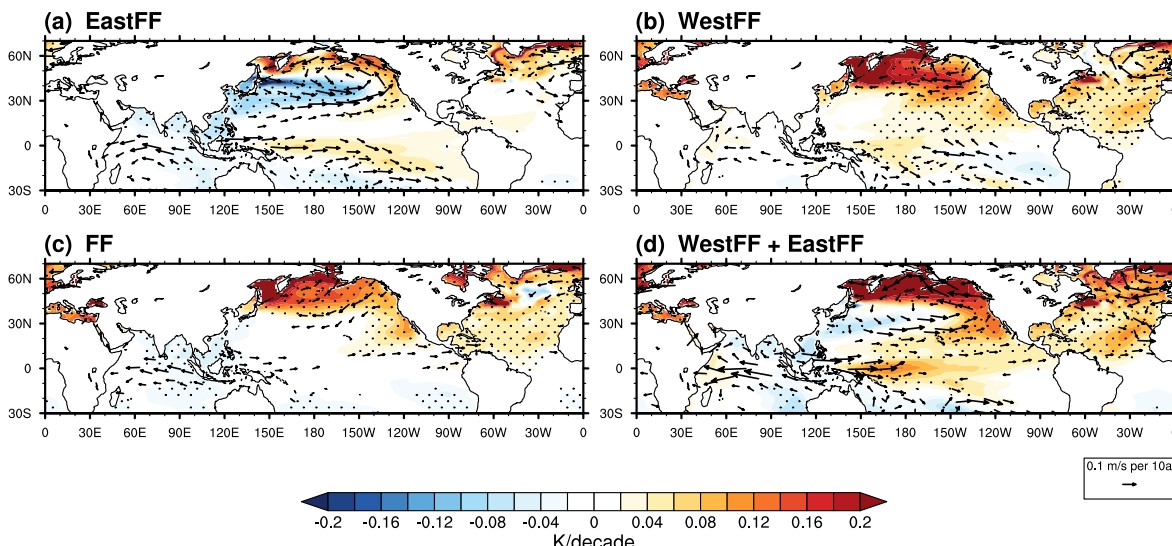

**Figure 2**

(a) Decadal changes in sea surface temperature (shading; K per decade) and 850 hPa horizontal wind (vectors; m s⁻¹ per decade) during 1980–2020 calculated in response to EastFF. (b) and (c): As in (a), but showing results for WestFF and FF, respectively. (d) Linear addition of panels (a) and (b). Stippled regions indicate significant values at the 90% confidence
level based on a two-sided t-test.

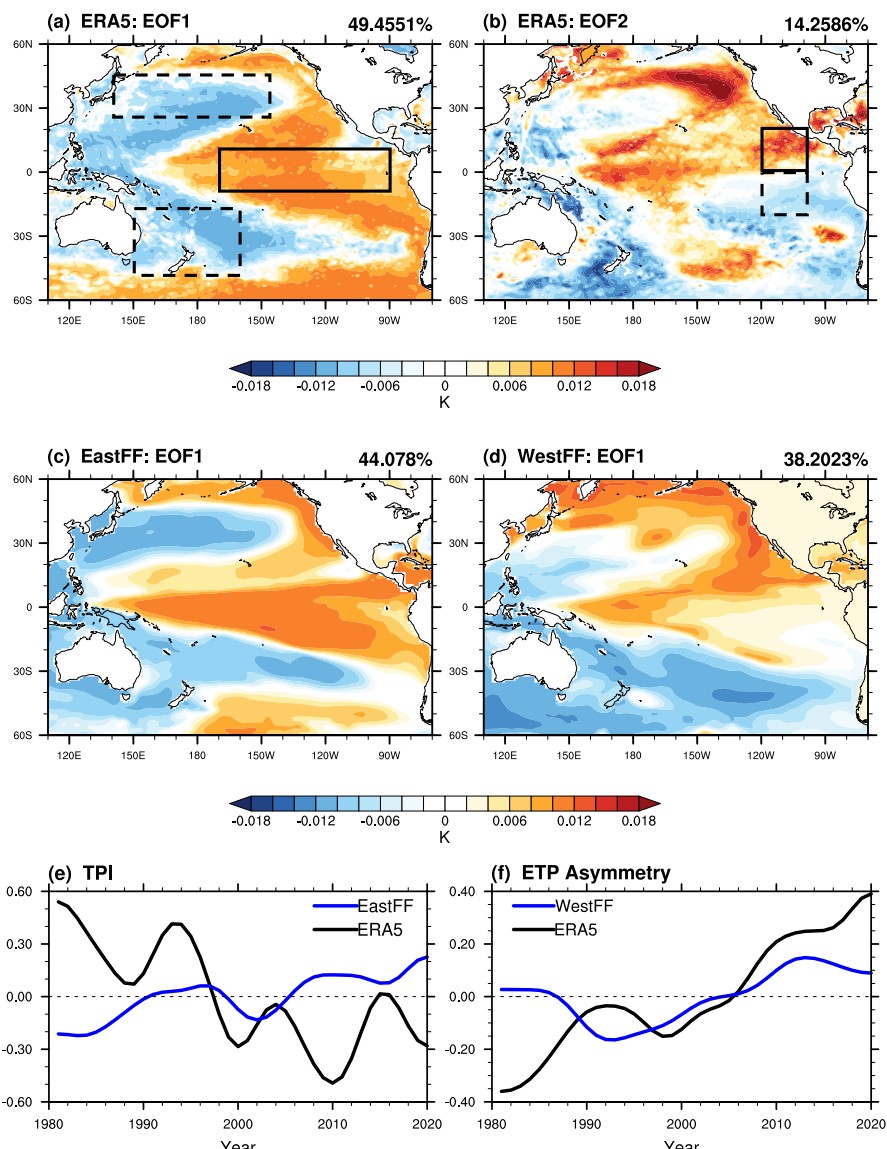

**Figure 3**

(a–b) The 1st and 2nd leading empirical orthogonal function (EOF) patterns of the observed SST anomalies from ERA5 over
the Pacific Ocean during 1980–2020. boxes in panel (a) represent the three key regions used to calculate the TPI index;
boxes in panel (b) represent the two key regions used to calculate the ETP asymmetry index (described in Sect. 2.3); (c) The
leading EOF pattern of the ensemble-mean SST responses from EastFF; (d) as in (c) but for WestFF result; (e) The time
series of TPI index obtained from (black) ERA5 and (blue) EastFF; (f) The time series of ETP asymmetry index obtained
from (black) ERA5 and (blue) WestFF.

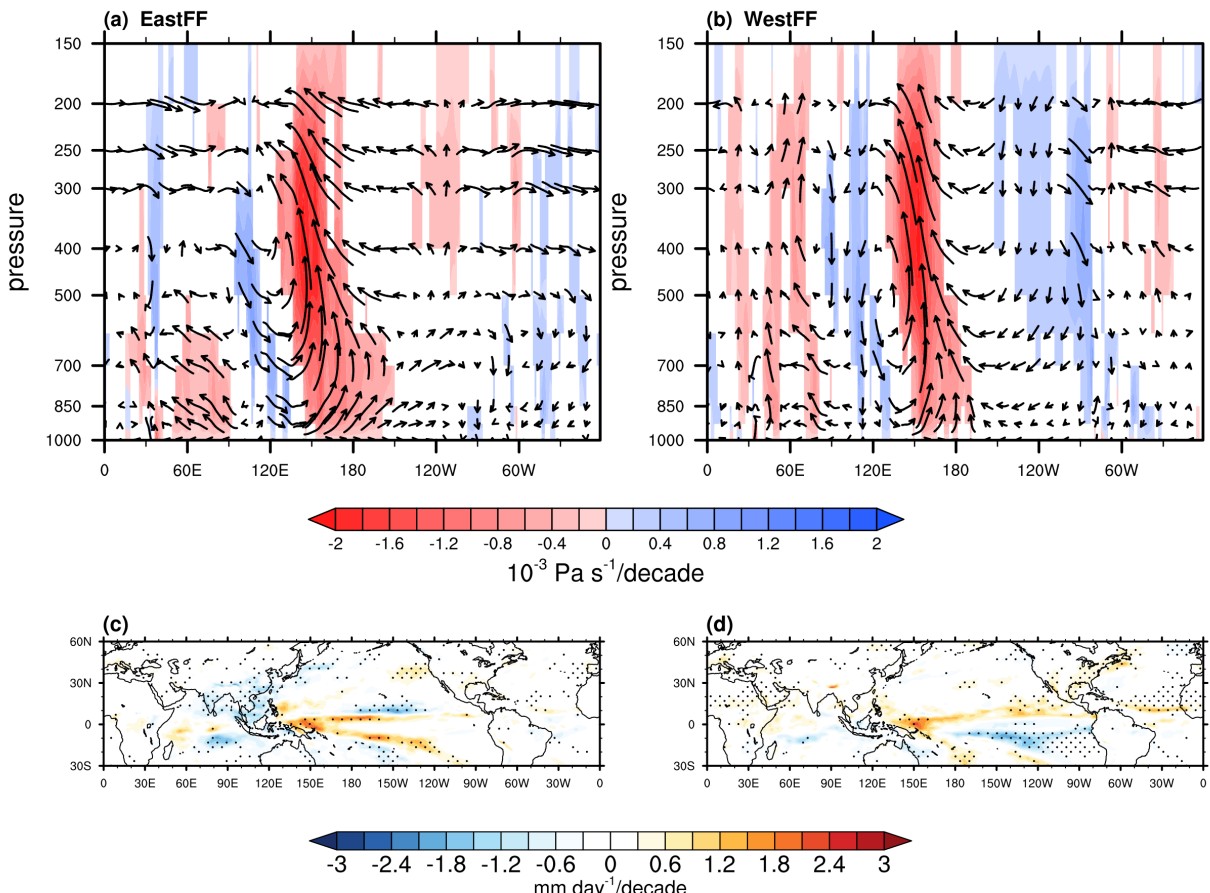

**Figure 4 Changes in the tropical circulation.**

(a) Decadal changes in cross-section of winds averaged from 5° S–5° N (vectors; The vertical component of the velocity vectors is scaled by a factor of 300) and vertical motion (shading; Pa s$^{-1}$ in response to EastFF. Blue shading indicates downward motions; red shading indicates upward motions). Regions that fail to pass the significance test (90% confidence level based on a two-sided t-test) are masked in white. (b) As in panel (a) but for WestFF. (c) Changes in tropical Precipitation (mm day$^{-1}$ per decade) in response to EastFF. (d) As in (c), but for WestFF. Stippled regions in (c) and (d) indicate significant values at the 90% confidence level based on a two-sided t-test.

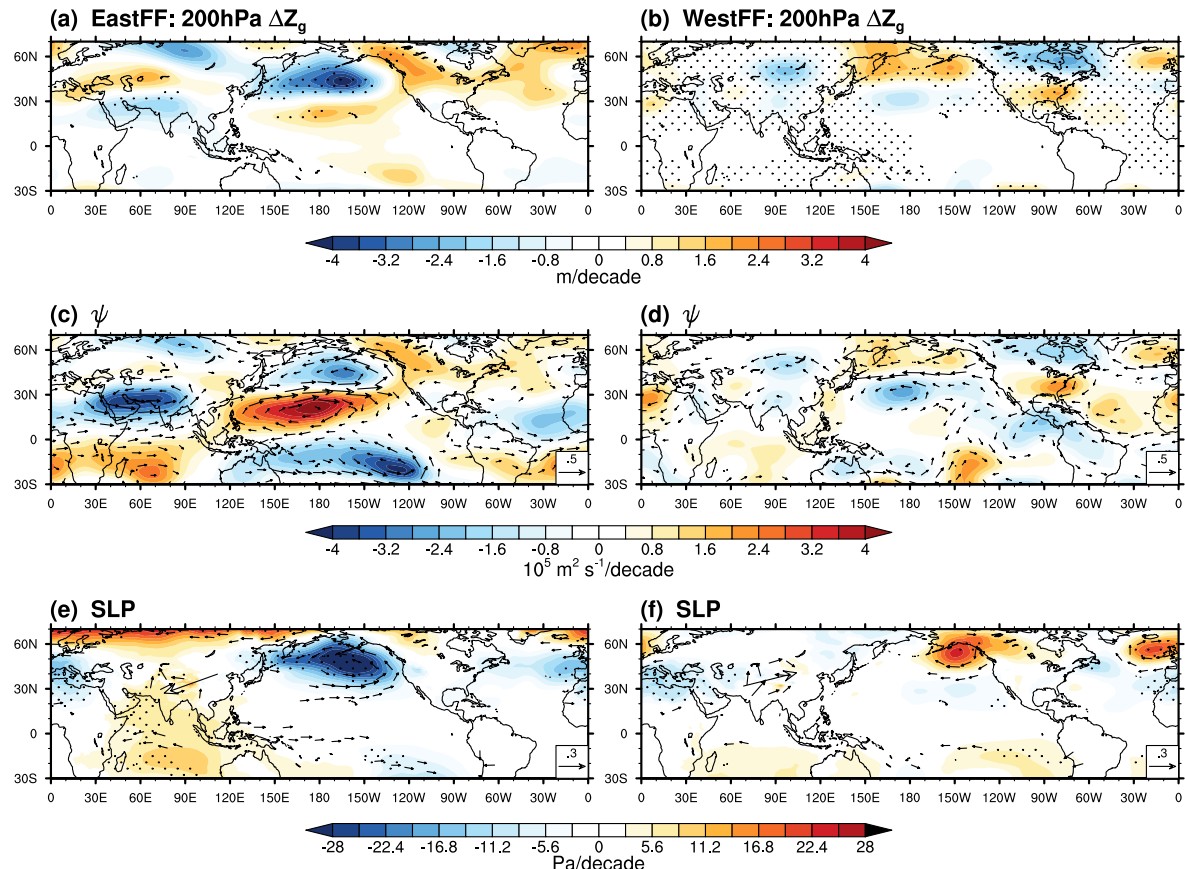

**Figure 5**

Left panels: EastFF-induced decadal changes of (a) 200 hPa eddy geopotential height (m per decade), (c) 250 hPa stream function (shading; m² s⁻¹ per decade), and wind (vectors; m s⁻¹ per decade), and (e) sea level pressure (shading; Pa per decade) and 850 hPa low-level wind (vectors; m s⁻¹ per decade). Right panels: same as Left panels, but due to WestFF.

Stippled regions indicate significant values at the 90% confidence level based on a two-sided t-test.

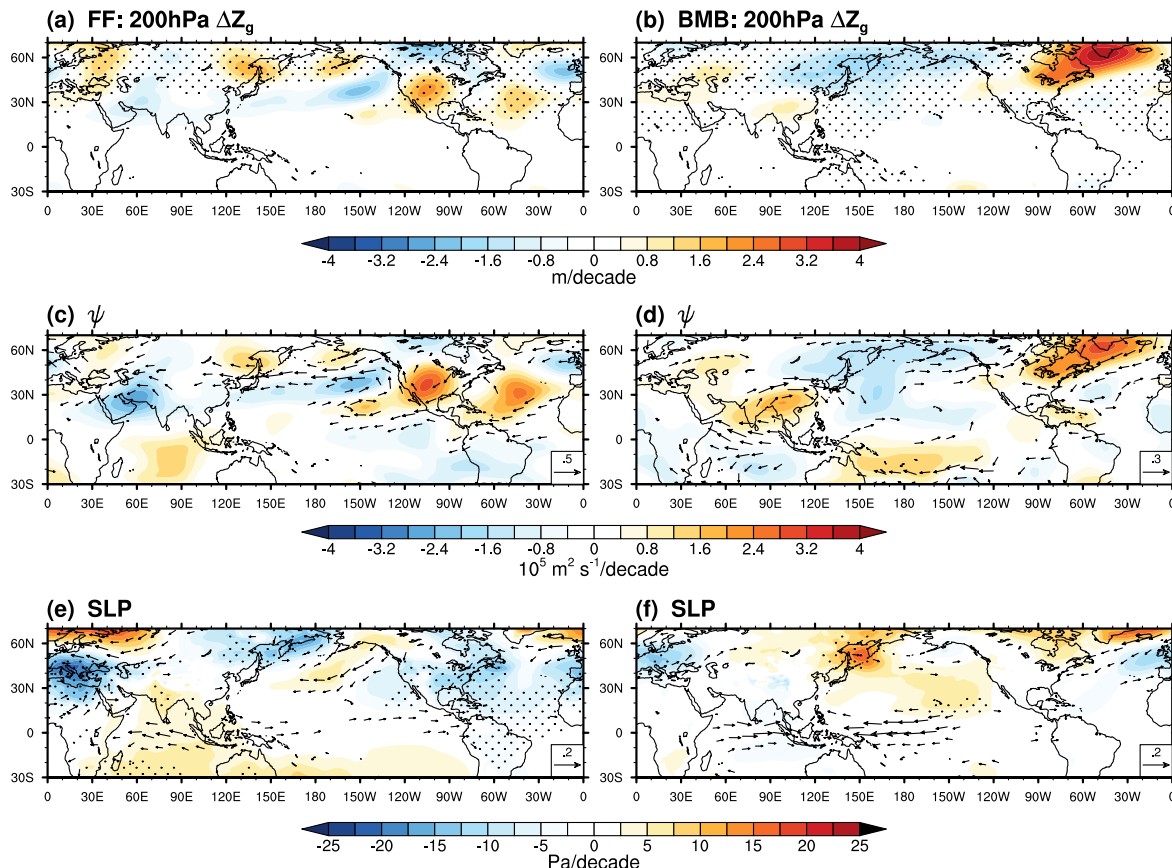

**Figure 6**

Left panels: FF-induced decadal changes (1980–2020) of (a) 200 hPa eddy geopotential height (m per decade), (c) 250 hPa
stream function (shading; m² s⁻¹ per decade), and wind. (vectors; m s⁻¹ per decade), and (e) sea level pressure (shading; Pa
per decade) and 850 hPa low-level wind (vectors; m s⁻¹ per decade). Right panels: same as Left panels, but due to Biomass
burning (BMB) simulations. Stippled regions indicate significant values at the 90% confidence level based on a two-sided t-
test.

## Code/Data availability

Datasets of the CESM1 Large Ensemble Project are available from: https://www.cesm.ucar.edu/projects/community-projects/LENS/data-sets.html (last access: 21 June 2024; Kay et al., 2015). Datasets of the CESM1 "Single Forcing" Large Ensemble Project are available from: https://www.cesm.ucar.edu/working_groups/CVC/simulations/cesm1-single_forcing_le.html (last access: 21 June 2024; Deser et al., 2020). Outputs for the two sets of regional single-forcing large-ensemble (Fix_EastFF1920 and Fix_WestFF1920; Diao et al., 2021) are available on the National Center for Atmospheric Research (NCAR) Campaign Storage file system and can also be accessed via the NCAR Data Sharing Service Endpoint on Globus upon request to the authors. All data analysis codes related to this study are available at: https://github.com/C-R-Diao/Climo_Pacific_reg-AA.git. At the time of publication, the codes will be converted to a permanent repository.

## Author contribution

CD and YX developed the idea for this study. CD performed the code development the data analyses, with input and feedback from YX, ZW, and AH. CD and YX prepared the paper, with contributions from all authors.

## Competing interests

The contact author has declared that neither they nor their co-authors have any competing interests.

## Acknowledgments

This work is supported by the National Science Foundation (NSF, grant no. 841308). We acknowledge NCAR for the high-performance computing support from Cheyenne (https://doi.org/10.5065/D6RX99HX) and the data storage resources provided by the Computational and Information Systems Laboratory (CISL), sponsored by the National Science Foundation. We thank the CESM-LE project and the CESM1 Single Forcing Large Ensemble Project for providing access to the model outputs.

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
