# Peer review of "Contrasting the roles of regional anthropogenic aerosols from the western and eastern Hemispheres in driving the 1980–2020 Pacific multi-decadal variations"

_EGUsphere, 2024_

## Author Comment (AC1)

**We would like to thank all three reviewers for their valuable comments and suggestions. We have responded to each referee's comment and revised the manuscript based on the suggestions. We also attached the modified figure at the end of this response for reference.**

**Referee #1**

**Summary**

The authors use a set of CESM1 simulations (1980-2020 transient runs; 10 members for each experiment) to isolate the impacts of fossil fuel/industrial aerosol emissions from China+India (EastFF; where emissions have increased) and North America+Europe (WestFF; where emissions have decreased) on the tropical Pacific and Pacific Decadal Variations (PDV). There are interesting differences between the two experiments, e.g., different North Pacific SST responses; EastFF drives a more El Nino-like SST pattern whereas WestFF drives a more CP-type El Nino SST pattern, etc. The authors continue onwards and discuss the dynamical mechanisms.

Overall, the paper is interesting and adds to our understanding of the potential climate impacts associated with regional changes in aerosol emissions.

**Response**

Thanks for the precise summary of the manuscript, which is mainly focused on the comparison of industrial aerosol forcings between West and East (WestFF vs. EastFF). We have revised the manuscript accordingly following the reviewer's comments and suggestions. Please check our detailed response below.

**Comments**

L23. "The competing effects of the heterogeneously distributed regional aerosol forcings are expected to be changed in the near future, which is likely to introduce opposite and more profound impacts of aerosol forcing on the Pacific multi-decadal changes." This is unclear. If

emissions decrease, this will reinforce the WestFF pattern showed here, but potentially flip the EastFF pattern showed here?  EastFF changes are likely complex, as emissions from China are likely to decrease (and have been decreasing, see below) whereas emissions from India may not, i.e., the east Asian dipole AOD pattern in Samset et al. 2019 (already referenced). Moreover, the components of EastFF and WestFF aerosol is different, as EastFF has a larger aerosol absorption component that has unique impacts on circulation, precipitation, etc. (e.g., https://www.nature.com/articles/nature11097).

**Response**

Thanks for pointing out this. We totally agree with the reviewer's comments on the complicated future changes in regional aerosol forcings. We also want to thank the reviewer for pointing out the difference in aerosol species, which was not covered in the previous manuscript.

We further clarified our statement starting at Line 23 (abstract) to be: "*The competing effects of the heterogeneously distributed regional aerosol forcings are expected to exhibit different patterns in the near future, especially the redistribution of aerosol emissions within the domain of EastFF (i.e., from East Asia to South Asia) and changes in aerosol composition. The complex future change in anthropogenic aerosol emissions is likely to introduce more profound impacts of aerosol forcing on the Pacific multi-decadal variations.*"

We also modified the related discussions about the aerosol composition in the last paragraph of the conclusion section:

"*In this study, we focus on the period of 1980–2020, when aerosol emissions over Asia show an overall increasing trend. However, studies have shown that the emission from East Asia reached a peak around the 2010s and started to decline recently while South Asia emission continues to increase (Samset et al., 2019; Ramachandran et al., 2020; Wang et al., 2021). The sensitivity of aerosol forcing's latitudinal placement within Asia is also highlighted by recent studies on dipole patterns emerging in Asia (Wang et al., 2022; Xiang et al., 2023). Also, the current offsetting effects between EastFF and WestFF can flip to a joint effect over specific regions of the North Pacific in the coming few decades. Such decline and/or redistribution of aerosol emissions can lead to distinct climate responses locally and remotely, which demands further continuous investigation in the future.*"

Linearity.  The authors show WestFF+EastFF relative to FF.  The FF AOD changes are well captured by WestFF+EastFF (Figure 1).  However, this is less true for the responses (e.g., Figure 2).  In particular, Figure 2 shows a much larger tropical Pacific response (the focus of this paper) under WestFF+EastFF as compared to FF.  Why might this be?  Is it related to aerosol outside these two regions?  I suppose this is unlikely given Figure 1.  Or is it related to differences between WestFF+EastFF versus West+EastFF (i.e., the latter is the signal from both perturbations simultaneously, which may be a better estimate for FF?).  I understand the authors have not performed the West+EastFF simulation, but some discussion on this matter is warranted.

**Response**

Thanks for the valuable comments. We agree with the reviewer that the nonlinearity is likely the primary reason for the difference between WestFF+EastFF and FF in the tropics. We add the significance test to the linear summation of WestFF+EastFF. Overall, most of the differences between FF and WestFF+EastFF in the tropical Pacific show statistically insignificant results, with some regional exceptions (e.g., Western Tropical Pacific). We now add more detailed discussions about the nonlinearity in the paragraph starting at line 194.

However, we still think the impact of aerosol forcings outside the two regions (i.e., in Africa and the Arabian Peninsula, see Fig. 1d) cannot be entirely ruled out, which could still partially contribute to the Walker circulation remotely, particularly by driving cooling over the tropical Indian Ocean region (Fig. 2d). Nevertheless, as mentioned by the reviewer, this impact is likely to be smaller than the nonlinearity, given the small magnitude of aerosol forcings.

We modified the paragraph starting at line 194 as follows:

*"The linear summation of WestFF and EastFF results presented in Fig. 2d shows greater SST responses in the tropical Pacific and tropical Indian Ocean compared with actual FF results (Fig. 2c). However, most of the differences between FF and the linear summation (warming signals) in the central and eastern tropical Pacific are statistically insignificant, with some regional exceptions. Notably, The equatorial West Pacific (160ºE–180º) exhibits a significant warming signal in WestFF+EastFF, which is likely due to the nonlinear interactions between the impacts of EastFF and WestFF. Similarly, in the*

*extratropical North Pacific, the linear summation closely resembled the EastFF signature while the actual FF results are dominated by WestFF. The Atlantic response appears to be largely consistent, with warming from both EastFF and WestFF. Note that the aerosol forcings outside the two focused regions (i.e., aerosols in Africa and the Arabian Peninsula; see Fig. 1d) could also partially contribute to the differences between FF and WestFF+EastFF, especially, particularly by driving cooling over the western tropical Indian Ocean and weakening the wind anomalies (Fig. 2c&d). Additionally, the aerosols outside the focused region could also impose a remote impact on the tropical Pacific region (Huang et al., 2021; Shi et al., 2022), but such impacts are likely to be smaller compared to the nonlinear interactions between EastFF and WestFF impacts, given the small magnitude of the forcing (Fig. 1d)."*

Related to the comments above is L93. "One note here is that the climate changes in response to FF do not necessarily equal the simple combination of that in response to EastFF and WestFF because the FF results also contain anthropogenic aerosol forcings originated from other regions not covered by EastFF and WestFF (e.g., Africa and Arabian Peninsula, Fig. 1d). More details of the regional AA single forcing large ensemble simulations are described in Diao et al. (2021)." This seems to suggest the differences in Figure 2c and Figure 2d are due to the very small differences in AOD from Figure 1d. Could there not be nonlinearities? Again, there are some sizable differences between Figure 2c and Figure 2d.

**Response**

Thanks for the great comment. We further clarify our statements here. Again, as mentioned above, we still want to keep a brief description of the aerosol outside focused regions, even though it might be small. The sentence now reads as follows:

*"One note here is that the climate changes in response to FF do not necessarily equal the simple sum of that in response to EastFF and WestFF (denoted as EastFF+WestFF) because FF can contain potential nonlinear interactions between EastFF and WestFF impacts. additionally, the FF results also contain aerosol forcings originating from other regions not covered by EastFF and WestFF (e.g., Africa and Arabian Peninsula, Fig. 1d), even though their magnitude is considerably smaller compared to the aerosol forcings in EastFF and WestFF."*

Biomass burning aerosols (BMB). Are they not important, i.e., in driving tropical Pacific SST variations? There is a very small discussion (e.g., Figure 6). A recent paper (https://www.nature.com/articles/s41612-024-00602-8) focused on the AMOC, using the same CESM1 Large Ensemble, and showed that BMB aerosols drove significant changes in the AMOC (that were largely out of phase relative to AMOC variations under FF aerosols).

**Response**

Thanks for the great comments. Yes, BMB also makes significant contributions to the Pacific decadal variations, particularly in the North Pacific based on the CESM1 single forcing large ensemble (see Fig. S1 below). However, in this study, we mainly focus on the climate impact of industrial aerosol forcings (FF), so we did not include detailed discussions on the BMB responses. The reason why we included BMB results in Fig. 6 is to highlight the importance of mid-latitude pathways driven by aerosol forcings located at high latitudes. We have now added a bit more descriptions and introductions of the BMB responses referring to existing studies and put a few related figures here in the response so as not to divert from the main focus of this manuscript.

The paragraph discussing Fig. 6 now reads as follows:

*"This sensitivity to the latitudinal displacement of forcing is supported by our secondary analysis of the biomass burning-related aerosol experiment (BMB), in which one major aerosol forcings are located in northeastern Asia in this particular model experiment (Fig. 2 in Diao et al., 2021), and we find similar wave trains propagating in the mid-latitudes (Fig. 6b). Similarly, in some other model experiments, the Atlantic heating, when placed in the extratropical, regardless of internal variability (Yao et al., 2021) or external forcings (Ruprich-Robert et al., 2017), does not excite the tropical teleconnection pathway. This further highlights the sensitive role of the latitudinal location of forcings. The BMB forcing excites a wave train propagating in the mid-latitudes, which later impacts the lower latitudes remotely. In fact, studies show that BMB, in addition to FF, also plays important roles in driving long-term climate variations (e.g., Fasullo et al., 2022; Tian et al., 2023; Yamaguchi et al., 2023; Allen et al., 2024), and more detailed analyses on the climate impacts of BMB on Pacific variations are warranted. However, since the focus of this study is the fossil fuel-related aerosol emissions, we leave such explorations to future work."*

A caveat/limitation of this study is the use of a single model, which should be acknowledged and discussed.  The results presented here are no doubt model dependent. The recently established Regional Aerosol Model Intercomparison Project (RAMIP) is one community effort designed to understand similar questions as addressed here (climate impacts of regional aerosol emissions changes) in a multi-model framework (https://gmd.copernicus.org/articles/16/4451/2023/gmd-16-4451-2023.html).

**Response**

Thanks for the suggestion. We agree with the reviewer's comments here. We further acknowledge the recent related work and add more discussions on the limitations of this work at the end the conclusion section. The new paragraph reads as follows:

*"Based on the single model large ensemble method, the simulations applied in this study effectively separate the externally forced climate responses from the model generated internal variations (Kay et al., 2015; Deser, et al., 2020; Diao et al., 2021). However, one limitation of this study is that all results shown in this study are purely based on a single climate model (i.e., CESM1), which inevitably includes model biases. Although CESM1 is proven to have good performance in aerosol simulations, it has a relatively larger aerosol effective radiative forcing among climate models ($-1.37\ W\ m^{-2}$ based on Deser et al., 2020). However, the recently established Regional Aerosol Model Intercomparison Project (RAMIP; Wilcox et al., 2023) introduces a new multi-model framework to explore the climate impacts of regional aerosols. Further analyses, similar to those covered in this study, based on the multi-model simulations in RAMIP are worth conducting to test the robustness of the conclusions presented here."*

Statistical significance and robustness of the ensemble mean signal.  This information is not provided in any figure.  We do not know what changes are significant, nor do we know the spread across individual realizations.  For example, maybe the multi-model mean response is driven by 1 or 2 ensemble members.  How robust are the responses shown here?

**Response**

Thanks for the constructive comments, we now added significance tests in Fig. 2, Fig. 4, Fig. 5, and Fig. 6. All our conclusions stay the same. Based on the significance test, we

updated the statements about the nonlinearity issues (see previous response). Also, we added the statistical significance statements here and there in the manuscript. Please check the modified figures by the end of the document.

Emissions. I assume CMIP5 emissions are used here?  And they are extended to 2020 using RCP4.5?  I just wonder about the similarities (or dissimilarities) between real-world changes in aerosol emissions and what is used to drive the model, particularly in the context of East Asia (largely China) emissions, where there are known disagreements.  For example, it is noted in a few places that EastFF is associated with "continuous cooling" (e.g., L140), which presumably means progressively more negative ERF and/or decreasing near-surface air temperatures regionally and/or globally?  Is this the case, or is there an inflection point where cooling transitions to warming (or perhaps the cooling levels off)?  If so, the EastFF forcing may be more complicated than is currently expressed.  Maybe this is not true in the modeling realm, but it might be true in the real-world, which certainly has impacts on one's ability to make/attempt attribution (e.g., Figure 3).

**Response**

Thanks for the comments. Yes, the experiments use the CMIP5 historical emission up to 2005 and then the RCP8.5 emission scenario thereafter (2005–2020). We followed this scenario setup to ensure our regional model experiments were comparable to the CESM1 single forcing large ensemble experiments.

Based on the observational studies (e.g., Wang et al., 2021; Xiang et al., 2023), aerosol emission in East Asia reached a peak and started to decrease slightly Since the early 2010s, while South Asia emissions continued to increase. Generally speaking, the RCP8.5 scenario overestimates the aerosol forcing in the EastFF domain from 2010 to 2020, which definitely introduces bias to the attribution in Fig. 3, as mentioned by the reviewer. Nevertheless, we want to mention that we only performed a *qualitative* comparison between the aerosol-driven multi-decadal responses and the observed variation. Also, although the aerosol forcing is overestimated in the simulations, the EastFF emission level remains at high levels from 2010 to 2020. Therefore, we believe that our comparison between model results and the observed

variations still indicates possible relationships between aerosol forcings and the observed results.

Following the reviewer's concern, we add caveat statements about the emission scenario issue at the end of Sect. 3.2 and also in the last paragraph of the conclusion section.

We add the following statement in Sect. 3.2:

*"One caveat to be noted is that the aerosol forcing scenario (RCP8.5) used in the Fix_EastFF1920 experiments has been proven to overestimate the aerosol emission level in East Asia since the early 2010s although it remains at a high level (Wang et al., 2021; Xiang et al., 2023). This leads to the overestimate of the EastFF forcing in the experiments. Therefore, the comparison above can only be treated as a qualitative comparison but not a quantitative attribution. In addition, the South Asia emission largely follows the emission scenario, which leads to a dipole of aerosol forcings changes within the EH. The forcing dipole might introduce complex circulation responses and new simulations with accurate emission forcings are necessary to further explore more realistic climate responses."*

We modified the last paragraph in the conclusion section:

*"In this study, we focus on the period of 1980–2020, when aerosol emissions over Asia show an overall increasing trend. However, studies have shown that the emission from East Asia reached a peak around the 2010s and have started to decline recently while South Asia emissions continue to increase (Samset et al., 2019; Ramachandran et al., 2020; Wang et al., 2021). The discrepancy between observed forcing and the forcing scenarios applied in the experiments in East Asia introduces additional biases to the results in EastFF, and new simulations with more accurate forcing scenarios are necessary to further test the climate response to regional aerosol forcings. In fact, the sensitivity of aerosol forcing's latitudinal placement within Asia is also highlighted by recent studies on dipole patterns emerging in Asia (Wang et al., 2022; Xiang et al., 2023). Moreover, the current offsetting effects between EastFF and WestFF can flip to a joint effect over specific regions of the North Pacific in the coming few decades. Such declines and/or redistribution of aerosol emissions can lead to distinctly complex climate responses locally and remotely, which demands further continuous investigation in the future."*

In a similar vein, additional details on the CESM1 model (specifically details relevant to this study) should probably be included. For example, CESM1 contains a relatively large aerosol ERF (this ties into the comment above on the fact this study uses one model).

**Response**

Thanks for the suggestion. We have included related caveat statements in the conclusion section. Please refer to our response above.

After Figure 2, plots generally only include EastFF and WestFF results (although Figure 6 throws in some BMB panels). What about FF? Do we not also want to compare EastFF and WestFF (or the linear sum of the two) to FF? One is left wondering if these two aerosol signals (or their sum) resemble the total FF signal. And more generally, what about the ALL forcing signal? Some discussion is perhaps warranted, e.g., does the ALL signal in any way look like FF and/or EastFF+WestFF? In other words, how important is the EastFF+WestFF signal relative to ALL/FF forcings. For example, Figure 5 and 6 attempt to do this, and it seems clear the FF dynamical response is quite different than BMB, as well as EastFF and WestFF (Fig. 5).

**Response**

Thanks for the comments. We only keep the results from EastFF and WestFF in order to focus on the different dynamical responses between EastFF and WestFF. Given that the FF and ALL responses have been widely discussed in previous studies (e.g., Smith et al., 2016; Deser et al., 2020), we are keen on highlighting the new results from the two regional forcing simulations.

Following the reviewer's suggestion, we now include a figure about the walker circulation response in FF in the supplementary document for comparison and add brief discussions in Sect. 3.3 as follows:

*"Similar to the SST changes, a strong nonlinear interaction between the impacts of EastFF and WestFF in the equatorial Pacific also occurs in FF results, where no significant PWC responses are exhibited in the Pacific (Fig. S1). Instead, FF induces a slight weakening of Walker circulations over the Indian Ocean and the Atlantic, which is consistent with the low-level wind anomalies shown in Fig. 2c."*

We also add brief discussions about the dynamcial responses from FF in Sect. 3.4 as follows:

*" The dynamical responses in FF largely follow the mid-latitude pathway in WestFF (Fig. 5a, e, &e), whereas the tropical Pacific shows no significant changes in Z200e. This is consistent with the surface patterns where WestFF dominates the North Pacific warming and tropical Pacific exhibits insignificant temperature changes (Fig. 2c). In addition, the North Atlantic in FF exhibits a significant decrease in sea level pressure, which is absent in either WestFF or EastFF. "*

Dynamical responses/teleconnections. The focus here is on annual means, but atmospheric teleconnections tend to have strong seasonal variations. Are these results (e.g., Fig. 5) largely boreal wintertime responses? A related paper that also addresses aerosol changes and their impacts on atmospheric circulation/teleconnections (but focused on Pacific Coast precipitation), which should probably be cited, is here:
https://iopscience.iop.org/article/10.1088/2752-5295/ac7d68/meta

**Response**

Thanks for the thoughtful question. We are not trying to touch the seasonality responses, but inspired by the reviewer, We conducted a simple analysis on the seasonality of wave teleconnections and found that the annual-mean dynamical responses are, as mentioned by the reviewer, largely consistent with the wintertime (DJF) wave teleconnections, but with a magnitude around one-third of that magnitude in wintertime. Because our study focuses only on the annual mean results throughout the manuscript and we do not want to include detailed analyses on the seasonal analyses, we would like to keep the annual mean results in the main text. However, we have included a brief discussion on the seasonality in Sect. 3.4. The discussion reads as follows:

*"Evidence also suggests that the teleconnection pathways induced by both EastFF and WestFF are predominantly associated with wintertime wave responses. Recent studies demonstrated that the impacts of anthropogenic aerosol forcings on regional precipitation are heterogeneous and seasonal dependent (e.g., Allen and Zhao 2022; Samset 2022). The distinct pathways of EastFF and WestFF are*

*likely to introduce complex regional precipitation patterns that vary seasonally, which warrants further investigation in future studies."*

[Figure]

***Fig. R1***

*Left panels: wintertime (DJF) dynamical response induced by EastFF (a) 200 hPa eddy geopotential height (m per decade), (c) 250 hPa stream function (shading; m2 s-1 per decade), and wind (vectors; m s-1 per decade), and (e) sea level pressure (shading; Pa per decade) and 850 hPa low-level wind (vectors; m s-1 per decade). Right panels: same as Left panels, but induced by WestFF. Stippled regions indicate insignificant values at the 90% confidence level based on a two-sided t-test.*

[Figure]

***Fig. R2***

*Left panels: wintertime (DJF) dynamical response induced by FF (a) 200 hPa eddy geopotential height (m per decade), (c) 250 hPa stream function (shading; m2 s-1 per decade), and wind (vectors; m s-1 per decade), and (e) sea level pressure (shading; Pa per decade) and 850 hPa low-level wind (vectors; m s-1 per decade). Right panels: same as Left panels, but induced by BMB. Stippled regions indicate insignificant values at the 90% confidence level based on a two-sided t-test.*

**Modified Figures**

[Figure]

*Figure 2*

*(a) Decadal changes in sea surface temperature (shading; K per decade) and 850 hPa horizontal wind (vectors; m s⁻¹ per decade) during 1980–2020 calculated in response to EastFF. (b) and (c): As in (a), but showing results for WestFF and FF, respectively. (d) Linear addition of panels (a) and (b). Stippled regions indicate insignificant values at the 90% confidence level based on a two-sided t-test.*

**Changes**

We add the significance test in all panels.

[Figure]

**Figure 4 Changes in the tropical circulation.**

*(a) Decadal changes in cross-section of winds averaged from 5° S–5° N (vectors; The vertical component of the velocity vectors is scaled by a factor of 300) and vertical motion (shading; Pa s-1 in response to EastFF. Blue shading indicates downward motions; red shading indicates upward motions). Regions that fail to pass the significance test (90% confidence level based on a two-sided t-test) are masked in white. (b) As in panel (a) but for WestFF. (c) Changes in tropical Precipitation (mm day-1 per decade) in response to EastFF. (d) As in (c), but for WestFF. Stippled regions in (c) and (d) indicate insignificant values at the 90% confidence level based on a two-sided t-test.*

**Changes**

We add the significance test.

[Figure]

*Figure 5*

*Left panels: EastFF-induced decadal changes of (a) 200 hPa eddy geopotential height (m per decade), (c) 250 hPa stream function (shading; m2 s-1 per decade), and wind (vectors; m s-1 per decade), and (e) sea level pressure (shading; Pa per decade) and 850 hPa low-level wind (vectors; m s-1 per decade). Right panels: same as Left panels, but due to WestFF. Stippled regions indicate insignificant values at the 90% confidence level based on a two-sided t-test.*

**Changes**

We add the significance test.

The colormap in panels e and f is reversed for better comparison, as suggested by the reviewer's suggestion.

[Figure]

*Figure 6*

*Left panels: FF-induced decadal changes (1980–2020) of (a) 200 hPa eddy geopotential height (m per decade), (c) 250 hPa stream function (shading; m² s⁻¹ per decade), and wind (vectors; m s⁻¹ per decade), and (e) sea level pressure (shading; Pa per decade) and 850 hPa low-level wind (vectors; m s⁻¹ per decade). Right panels: same as Left panels, but due to Biomass burning (BMB) simulations. Stippled regions indicate insignificant values at the 90% confidence level based on a two-sided t test.*

**Changes**

We add the significance test.

The colormap in panels e and f is reversed for better comparison, as suggested by the reviewer's suggestion.

[Figure]

***Figure S1 (new figure)***

*So as Fig. 4 but for (left) FF, and (right) BMB responses.*

---

## Author Comment (AC2)

**We would like to thank all three reviewers for their valuable comments and suggestions. We have responded to each referee's comment and revised the manuscript based on the suggestions. We also attached the modified figure at the end of this response for reference.**

**Referee #2**

Since the 1980s, the anthropogenic aerosols (AA) in the western hemisphere are reduced whereas those in the eastern hemisphere continue to increase. The study is to contrast the effect of regional AA from the western and eastern hemispheres in driving the Pacific climate change in the past four decades, using large ensemble regional AA forcing simulations. The analysis is straightforward and the manuscript is clearly written. I only have some minor comments.

**Response**

Thanks for the inspiring review and good assessment of our manuscript. Please check our responses below.

1.      Wang et al. (2024) show that the AA in China has been decreasing between 2007 and 2020. Is this also reflected in your experiment setup? Or is it assumed to have continuously been increasing? Uncertainties in the AA forcing make me question the validity of comparing regional AA simulations with observations. Caveats regarding the uncertainties in the AA forcing should be discussed.

**Response**

Thanks for the comments. Yes, the experiments use the CMIP5 historical emission up to 2005 and then the RCP8.5 emission scenario thereafter (2005–2020). We followed this scenario setup to ensure our regional model experiments were comparable to the CESM1 single forcing large ensemble experiments.

Based on the observational studies (e.g., Wang et al., 2021; Xiang et al., 2023), aerosol emission in East Asia reached a peak and started to decrease slightly Since the early 2010s,

while South Asia emissions continued to increase. Generally speaking, the RCP8.5 scenario overestimates the aerosol forcing in the EastFF domain from 2010 to 2020, which definitely introduces bias to the attribution in Fig. 3, as mentioned by the reviewer. Nevertheless, we want to mention that we only performed a *qualitative* comparison between the aerosol-driven multi-decadal responses and the observed variation. Also, although the aerosol forcing is overestimated in the simulations, the EastFF emission level remains at high levels from 2010 to 2020. Therefore, we believe that our comparison between model results and the observed variations still indicates possible relationships between aerosol forcings and the observed results.

Following the reviewer's concern, we add caveat statements about the emission scenario issue at the end of Sect. 3.2 and also in the last paragraph of the conclusion section.

We add the following statement in Sect. 3.2:

*"One caveat to be noted is that the aerosol forcing scenario (RCP8.5) used in the Fix_EastFF1920 experiments has been proven to overestimate the aerosol emission level in East Asia since the early 2010s although it remains at a high level (Wang et al., 2021; Xiang et al., 2023). This leads to the overestimation of the EastFF forcing in the experiments. Therefore, the comparison above can only be treated as a qualitative comparison but not a quantitative attribution. In addition, the South Asia emission largely follows the emission scenario, which leads to a dipole of aerosol forcings changes within the EH. The forcing dipole might introduce complex circulation responses and new simulations with accurate emission forcings are necessary to further explore more realistic climate responses."*

We modified the last paragraph in the conclusion section:

*"In this study, we focus on the period of 1980–2020, when aerosol emissions over Asia show an overall increasing trend. However, studies have shown that the emission from East Asia reached a peak around the 2010s and have started to decline recently while South Asia emissions continue to increase (Samset et al., 2019; Ramachandran et al., 2020; Wang et al., 2021). The discrepancy between observed forcing and the forcing scenarios applied in the experiments in East Asia introduces additional biases to the results in EastFF, and new simulations with more accurate forcing scenarios are necessary to further test the climate response to regional aerosol forcings. In fact, the sensitivity of aerosol forcing's latitudinal placement within Asia is also highlighted by recent studies on dipole patterns emerging in Asia (Wang et al., 2022; Xiang et al., 2023). Moreover, the current offsetting effects between EastFF and*

*WestFF can flip to a joint effect over specific regions of the North Pacific in the coming few decades. Such declines and/or redistribution of aerosol emissions can lead to distinctly complex climate responses locally and remotely, which demands further continuous investigation in the future."*

2.    Indicate the significance of the responses in the figures.

**Response**

  Thanks for the constructive comments, we now added significance tests in Fig. 2, Fig. 4, Fig. 5, and Fig. 6. All our conclusions stay the same. Based on the significance test, we updated the statements about the nonlinearity issues (see previous response). Also, we added the statistical significance statements here and there in the manuscript. Please check the modified figures by the end of the document.

3.    Section 3.2: I presume the EOF is applied to the ensemble-mean. What about applying the EOF to each ensemble, and then taking the mean? It is more comparable to contrast the ERA5 with each ensemble rather than with ensemble-mean. It'll also be useful to see the spread of indices in Fig. 3e,f.

**Response**

  Thanks for the suggestion. However, with respect, we do not think the mean of EOF for each ensemble makes sense here. First, the 1st and 2nd EOF patterns of each ensemble member should represent the global warming and randomly modeled internal variability, because these two have much larger signals in single realizations compared to aerosol forcings. This is why we performed large ensemble simulations to separate the aerosol forcings from the internal variability. Therefore, direct comparisons between ERA5 and single ensemble members cannot clearly show the climate responses to aerosol forcings from our understanding. Second, our forced response is calculated by subtracting "fixed regional aerosol ensemble mean" from "all forcing ensemble mean", if we want to generate EOF for each ensemble member, we have to randomly pick one realization from the two experiments and calculate the difference, which introduces subtraction between two random generated internal variability. Hence, we would like to keep our figures to EOF to ensemble-mean.

4. The authors explain the upper-level low anomaly over the extratropial North Pacific in EastFF as a result of a Rossby wave train excited from the enhanced convection over the central equatorial Pacific (Fig. 5a,c,e).

**Response**

Thanks for the good assessment of our paper.

The precipitation response in the tropical Pacific exhibits a comparable order of magnitude between EastFF and WestFF (Fig. 4c,d). In particular, both simulations show precipitation increase in the western tropical Pacific. I'd expect a similar Rossby wave pattern driven by the enhanced diabatic heating over the warm pool. However, the response of upper-level wave activity is much weaker in WestFF. The fact that the tropical precipitation response is similar between EastFF and WestFF makes me think that strong circulation responses in the extratropical Pacific in EastFF are due to local AA radiative forcing via modulating storm tracks. That said, I don't follow the discussion in Section 3.4.

**Response**

Thanks for the comments. Based on our results, the precipitation responses in EastFF and WestFF show different mechanisms: The EastFF, which has forcings at low latitudes near the warm pool region, excites the typical El Niño-like SST and precipitation patterns in the tropical Pacific, which serves as the wave source for tropical-extratropical teleconnection. Meanwhile, the WestFF, which has forcings at mid-to-high latitudes, excites the teleconnection from mid-latitudes to the tropics and remotely impacts the tropical regions. Therefore, the tropical rainfall in WestFF is the climate response but not the wave source. The storm track analyses in an interesting idea to further explore the extratropical precipitation response. However, given the small extratropical precipitation responses and the weak local aerosol forcings, we think it is beyond the scope of the current study and decided not to include this analysis in this manuscript.

**Modified Figures**

[Figure]

*Figure 2*

*(a) Decadal changes in sea surface temperature (shading; K per decade) and 850 hPa horizontal wind (vectors; m s$^{-1}$ per decade) during 1980–2020 calculated in response to EastFF. (b) and (c): As in (a), but showing results for WestFF and FF, respectively. (d) Linear addition of panels (a) and (b). Stippled regions indicate insignificant values at the 90% confidence level based on a two-sided t-test.*

**Changes**

We add the significance test in all panels.

[Figure]

**Figure 4 Changes in the tropical circulation.**

*(a) Decadal changes in cross-section of winds averaged from 5° S–5° N (vectors; The vertical component of the velocity vectors is scaled by a factor of 300) and vertical motion (shading; Pa s-1 in response to EastFF. Blue shading indicates downward motions; red shading indicates upward motions). Regions that fail to pass the significance test (90% confidence level based on a two-sided t test) are masked in white. (b) As in panel (a) but for WestFF. (c) Changes in tropical Precipitation (mm day-1 per decade) in response to EastFF. (d) As in (c), but for WestFF. Stippled regions in (c) and (d) indicate insignificant values at the 90% confidence level based on a two-sided t-test.*

**Changes**

We add the significance test.

[Figure]

*Figure 5*

*Left panels: EastFF-induced decadal changes of (a) 200 hPa eddy geopotential height (m per decade), (c) 250 hPa stream function (shading; m2 s-1 per decade), and wind (vectors; m s-1 per decade), and (e) sea level pressure (shading; Pa per decade) and 850 hPa low-level wind (vectors; m s-1 per decade). Right panels: same as Left panels, but due to WestFF. Stippled regions indicate insignificant values at the 90% confidence level based on a two-sided t-test.*

**Changes**

We add the significance test.

The colormap in panels e and f is reversed for better comparison, as suggested by reviewer's suggestion.

[Figure]

*Figure 6*

*Left panels: FF-induced decadal changes (1980–2020) of (a) 200 hPa eddy geopotential height (m per decade), (c) 250 hPa stream function (shading; m² s⁻¹ per decade), and wind (vectors; m s⁻¹ per decade), and (e) sea level pressure (shading; Pa per decade) and 850 hPa low-level wind (vectors; m s⁻¹ per decade). Right panels: same as Left panels, but due to Biomass burning (BMB) simulations. Stippled regions indicate insignificant values at the 90% confidence level based on a two-sided t-test.*

**Changes**

We add the significance test.

The colormap in panels e and f is reversed for better comparison, as suggested by reviewer's suggestion.

[Figure]

***Figure S1 (new figure)***

*So as Fig. 4 but for (left) FF, and (right) BMB responses.*

---

## Author Comment (AC3)

**We would like to thank all three reviewers for their valuable comments and suggestions. We have responded to each referee's comment and revised the manuscript based on the suggestions. We also attached the modified figure at the end of this response for reference.**

**Referee #3**

**General comments**

By using regional aerosol forcing large ensemble simulations based on CESM1 performed earlier, this study investigates impacts of increasing fossil fuel-related aerosol emission over Asia (EastFF) and the reduction in aerosol emission over North America and Europe (WestFF) on the Pacific circulations and SST changes since the 1980s.

**Response**

Thanks for the precise summary of the manuscript, which is mainly focused on the comparison of industrial aerosol forcings between West and East (WestFF vs. EastFF). We have revised the manuscript accordingly following the reviewer's comments and suggestions. Please check our detailed response below.

One major concern the reviewer has is that results presented in the current version of manuscript are lack of significant test. It is not clear whether responses to different regional aerosols forcing changes described in the study are statistically significant from the internal variability. In addition, there are some other comments that need to be addressed to improve the manuscript. Therefore, the paper needs a major revision before it can be considered for publication.

**Response**

Thanks for pointing out this important question. We have added significance test results in Fig. 2, Fig. 4, Fig. 5, and Fig. 6. All our conclusions stay convincing. Based on the significance test, we updated the statements about the nonlinearity issues. Also, we added the statistical significance statements here and there in the manuscript. Please check the modified figures by the end of the document.

**Major comments**

1.      It is not clear whether all results analysed are about annual changes. In data and method, there is no any information given.

**Response**

Thanks for pointing out this issue. All analyses are based on the annual mean results. We clarify this in the Method section as follows:

*"All analyses in this study are based on the ensemble-averaged results of the monthly outputs of the five experiments mentioned above (ALL, FF, EastFF, WestFF, and BMB) to exclude the impact of randomly generated internal variability in the model. Annual means are calculated prior to analyses."*

2.      All results presented in the current version of manuscript are lack of significant test. It is not clear whether responses described are statistically significant from the internal variability. This aspect needs to be improved through whole manuscript.

**Response**

Thanks for pointing out this important question. We have added significance test results in Fig. 2, Fig. 4, Fig. 5, and Fig. 6. All our conclusions stay convincing. Based on the significance test, we updated the statements about the nonlinearity issues. Also, we added the statistical significance statements here and there in the manuscript. Please check the modified figures by the end of the document.

3.      Many references cited in text are not in reference list. Please check them carefully.

**Response**

We really appreciate the careful review. We have updated the reference list.

4.    The study is based on a set of single model simulations, some comments on this aspect would be helpful for readers.

**Response**

Thanks for the suggestion. We further acknowledge the recent related work and add more discussions on the caveat and limitation of the single model results at the end the conclusion section. The new paragraph reads as follows:

*"Based on the single model large ensemble method, the simulations applied in this study effectively separate the externally forced climate responses from the model-generated internal variations (Kay et al., 2015; Deser, et al., 2020; Diao et al., 2021). However, one limitation of this study is that all results shown in this study are purely based on a single climate model (i.e., CESM1), which inevitably includes model biases. Although CESM1 is proven to have good performance of aerosol simulations, it has a relatively larger aerosol effective radiative forcing among climate models ($-1.37$ W m$^{-2}$ based on Deser et al., 2020). However, the recently established Regional Aerosol Model Intercomparison Project (RAMIP; Wilcox et al., 2023) introduces a new multi-model framework to explore the climate impacts of regional aerosols. Further analyses, similar to those covered in this study, based on the multi-model simulations in RAMIP are worth conducting to test the robustness of the conclusions presented here."*

Specific comments

1.    Line 20. "an IPO-like SST pattern". It is helpful if authors can clarify the phase.

Response

Thanks for the suggestion. We have modified this to be " *an IPO-like SST pattern (horseshoe-like SST pattern in the North Pacific)*" in the abstract, which is more commonly described in previous studies. We also clarified this in Sect. 3.1.

2.    Lines 23-25. The last sentence is very confusing. How the likely opposite responses to future aerosol emission changes are likely to introduce more profound impacts?

Response

Thanks for pointing out this issue. We completely rewrite the statements here following suggestions from another reviewer. The sentences now read as follows:

*"The competing effects of the heterogeneously distributed regional aerosol forcings are expected to exhibit different patterns in the near future, especially the redistribution of aerosol emissions within the domain of EastFF (i.e., from East Asia to South Asia) and changes in aerosol composition. The complex future changes in anthropogenic aerosol emissions are likely to introduce more profound impacts of aerosol forcing on the Pacific multi-decadal variations."*

3.      Lines 112-113. Are SST anomalies low frequency filtered? Please clarify.

**Response**

Thanks for the comment. We applied an 11-year low-pass filter before EOF to obtain the interdecadal variability. We add the description at line 113 as follows:

*"An 11-year low-pass filter is applied to the SST anomalies prior to the EOF analyses in order to obtain the interdecadal variability."*

4.      Line 115. "global warming mode induced by the greenhouse gases (GHG)". There are also other external forcings in model simulations.

**Response**

Thanks for the correction. We modified the sentence as follows:

*"For the model simulation results, we subtract the global averaged SST time series from the simulated SST patterns before EOF analyses in order to remove the global change modes driven by external forcings."*

5.      Lines 195-197. Do authors really think that the discrepancy between FF response and sum of EastFF and WestFF responses is due to aerosol forcings outside two focused regions, which are small as suggested in Figure 1d?

**Response**

Thanks for the valuable comments. We argue that the nonlinearity is likely the primary reason for the difference between WestFF+EastFF and FF in the tropics. We add the significance test to the linear summation of WestFF+EastFF. Overall, most of the differences between FF and WestFF+EastFF in the central and eastern tropical Pacific show statistically insignificant results, with some regional exceptions (e.g., Western Tropical Pacific). We now add more detailed discussions about the nonlinearity in the paragraph starting at line 194.

However, we still think the impact of aerosol forcings outside the two regions (i.e., in Africa and the Arabian Peninsula, see Fig. 1d) cannot be entirely ruled out, which could still partially contribute to the Walker circulation remotely, particularly by driving cooling over the tropical Indian Ocean region (Fig. 2d). Nevertheless, as mentioned by the reviewer, this impact is likely to be smaller than the nonlinearity, given the small magnitude of aerosol forcings.

We modified the paragraph starting at line 194 as follows:

*"The linear summation of WestFF and EastFF results presented in Fig. 2d shows greater SST responses in the tropical Pacific and tropical Indian Ocean compared with actual FF results (Fig. 2c). However, most of the differences between FF and the linear summation (warming signals) in the central and eastern tropical Pacific are statistically insignificant, with some regional exceptions. Notably, The equatorial West Pacific (160ºE–180º) exhibits a significant warming signal in WestFF+EastFF, which is likely due to the nonlinear interactions between the impacts of EastFF and WestFF. Similarly, in the extratropical North Pacific, the linear summation closely resembled the EastFF signature while the actual FF results are dominated by WestFF. The Atlantic response appears to be largely consistent, with warming from both EastFF and WestFF. Note that the aerosol forcings outside the two focused regions (i.e., aerosols in Africa and the Arabian Peninsula; see Fig. 1d) could also partially contribute to the differences between FF and WestFF+EastFF, especially, particularly by driving cooling over the western tropical Indian Ocean and weakening the wind anomalies (Fig. 2c&d). Additionally, the aerosols outside the focused region could also impose a remote impact on the tropical Pacific region (Huang et al., 2021; Shi et al., 2022), but such impacts are likely to be smaller compared to the nonlinear interactions between EastFF and WestFF impacts, given the small magnitude of the forcing (Fig. 1d)."*

6.      Line 204. "SST anomalies". See major comment 1 and specific comment 3.

**Response**

Thanks for the comment. We have clarified the method of EOF analyses in the Method section.

7.      Lines 254-256. These statements about changes in Walker circulation over the equatorial Atlantic are not convincing. Figure 4b shows anomalous ascent in the tropical Atlantic, being consistent with warming in the tropical Atlantic (Fig. 2b). However, it is hard to argue whether this change is statically significant or not since no such test is shown. See major comment 2.

**Response**

Thanks for the comments. We have added the significance test results to all related figures. Please check the modified figure at the end of this document.

To clarify here, we wanted to mention that WestFF does not induce any noticeable Walker circulation changes over the equatorial Atlantic, although Fig. 2b shows a warming trend over the tropical Atlantic. Indeed, the ascent in the equatorial Atlantic is insignificant at the lower levels (check modified Fig. 2).

8.      Lines 278-279. (Fig. 5c). Shall be (Fig. 5e)?

**Response**

Thanks for the correction. Yes, it should be Fig. 5e. We have corrected this in the manuscript.

9.      Line 288. "convection", use another word.

**Response**

Thanks for the suggestion. We have changed it to be "upper-level geopotential height pattern".

10.     Lines 298-303. These arguments about weak tropical teleconnection are not very convincing. See major comment 2 and specific comment 7.

**Response**

Thanks for the comment. We have added the significance test results to Fig. 5. Based on the significance test and the results of the BMB experiment and other extratropical forcing experiments as mentioned in the manuscript, we argue that our discussion about the mid-latitude pathway is the key difference between the WestFF and EastFF response.

11.     Lines 305-310. It is not clear what are the aims for showing BMB experiment in Fig. 6b, d, f. What wave trains authors describe here and they are similar to what?

**Response**

Thanks for the great comments. The reason why we included BMB results in Fig. 6 is to highlight the importance of mid-latitude pathways driven by aerosol forcings located at high latitudes. We also mentioned some other extratropical forcing experiments from previous studies to further indicate the importance of this mechanism. We have now added a bit more descriptions and introductions of the BMB responses referring to existing studies.  We now include Fig. 6 to the supplementary document so as not to divert from the main focus of this manuscript – a comparison between EastFF and WestFF.

The paragraph discussing the BMB results now reads as follows:

*"This sensitivity to the latitudinal displacement of forcing is supported by our secondary analysis of the biomass burning-related aerosol experiment (BMB), in which one major aerosol forcings are located in northeastern Asia in this particular model experiment (Fig. 2 in Diao et al., 2021), and we find similar wave trains propagating in the mid-latitudes (Fig. 6b). Similarly, in some other model experiments, the Atlantic heating, when placed in the extratropical, regardless of internal variability (Yao et al., 2021) or external forcings (Ruprich-Robert et al., 2017), does not excite the tropical teleconnection pathway. This further highlights the sensitive role of the latitudinal location of forcings. The BMB forcing excites a wave train propagating in the mid-latitudes, which later impacts the lower latitudes remotely. In fact, studies show that BMB, in addition to FF, also plays important roles in driving long-term climate variations (e.g., Fasullo et al., 2022; Tian et al., 2023; Yamaguchi et al.,*

*2023; Allen et al., 2024), and more detailed analyses on the climate impacts of BMB on Pacific variations are warranted. However, since the focus of this study is the fossil fuel-related aerosol emissions, we leave such explorations to future work."*

Fig. 6a, c, e are not refereed in text.

**Response**

Thanks for pointing out this. We now added additional brief descriptions of the dynamical responses to FF in Sect. 3.4, where we cited the Fig. 6a, c, &e. It reads as follows:

*"The dynamical responses in FF largely follow the mid-latitude pathway in WestFF (Fig. 6a, e, &e), whereas the tropical Pacific shows no significant changes in Z200e. This is consistent with the surface patterns where WestFF dominates the North Pacific warming and tropical Pacific exhibits insignificant temperature changes (Fig. 2c). In addition, the North Atlantic in FF exhibits a significant decrease in sea level pressure, which is absent in either WestFF or EastFF."*

12.    Lines 342-344. See major comment 2, and specific comments 7 and 10.

**Response**

Thanks for the comments. Based on the significance test results, we argue that our discussions here are convincing. Please check our previous responses.

13.    Lines 346-348. See specific comment 11.

**Response**

Thanks for the comment. Please check our response about the BMB results above.

14.    Figure 5e, f and Figure 6e, f. Reverse colour scale for easy comparison with other panels.

**Response**

Thanks for the suggestion. We now reversed the color scale for Fig. 5e& f and Fig. 6e&f.

**Modified Figures**

[Figure]

*Figure 2*

*(a) Decadal changes in sea surface temperature (shading; K per decade) and 850 hPa horizontal wind (vectors; m s⁻¹ per decade) during 1980–2020 calculated in response to EastFF. (b) and (c): As in (a), but showing results for WestFF and FF, respectively. (d) Linear addition of panels (a) and (b). Stippled regions indicate insignificant values at the 90% confidence level based on a two-sided t-test.*

**Changes**

We add the significance test in all panels.

[Figure]

*Figure 4 Changes in the tropical circulation.*

*(a) Decadal changes in cross-section of winds averaged from 5° S–5° N (vectors; The vertical component of the velocity vectors is scaled by a factor of 300) and vertical motion (shading; Pa s-1 in response to EastFF. Blue shading indicates downward motions; red shading indicates upward motions). Regions that fail to pass the significance test (90% confidence level based on a two-sided t-test) are masked in white. (b) As in panel (a) but for WestFF. (c) Changes in tropical Precipitation (mm day-1 per decade) in response to EastFF. (d) As in (c), but for WestFF. Stippled regions in (c) and (d) indicate insignificant values at the 90% confidence level based on a two-sided t-test.*

**Changes**

We add the significance test.

[Figure]

*Figure 5*

*Left panels: EastFF-induced decadal changes of (a) 200 hPa eddy geopotential height (m per decade), (c) 250 hPa stream function (shading; m2 s-1 per decade), and wind (vectors; m s-1 per decade), and (e) sea level pressure (shading; Pa per decade) and 850 hPa low-level wind (vectors; m s-1 per decade). Right panels: same as Left panels, but due to WestFF. Stippled regions indicate insignificant values at the 90% confidence level based on a two-sided t-test.*

**Changes**

We add the significance test.

The colormap in panels e and f is reversed for better comparison, as suggested by reviewer's suggestion.

[Figure]

***Figure 6***

*Left panels: FF-induced decadal changes (1980–2020) of (a) 200 hPa eddy geopotential height (m per decade), (c) 250 hPa stream function (shading; m² s⁻¹ per decade), and wind (vectors; m s⁻¹ per decade), and (e) sea level pressure (shading; Pa per decade) and 850 hPa low-level wind (vectors; m s⁻¹ per decade). Right panels: same as Left panels, but due to Biomass burning (BMB) simulations. Stippled regions indicate insignificant values at the 90% confidence level based on a two-sided t-test.*

**Changes**

We add the significance test.

The colormap in panels e and f is reversed for better comparison, as suggested by reviewer's suggestion.

[Figure]

***Figure S1 (new figure)***

*So as Fig. 4 but for (left) FF, and (right) BMB responses.*

---

## Author Response (AR2)

Thank you for your efforts to address the reviewers' comments. I want to raise just a couple of points where I think there are remaining issues from the discussion.

**Response:**

We really appreciate your careful read and great comments/suggestions. We have responded to each comment and revised the manuscript accordingly.

The question of linearity was raised by the referees, and it is good to see that you have included some discussion of this in the revised manuscript. However, while you note that there are some clear nonlinearities present in response, you conclude that they are largely insignificant. I have some concerns about this conclusion, as I think there may be an error in the interpretation of the significance test. In all figures with stippling, the caption says that stippling indicates where the response is insignificant. However, the stippling in all figures covers the largest responses, and is absent from the regions with no change. Please can you double check whether your stippling is indicating significant or insignificant responses, and update your discussion accordingly (and update the discussion around the nonlinearities in the response in particular).

**Response:**

Thanks for the comments.

After checking the figure captions and codes, we found that it was a typo—the stippling in all figures shows the regions with _significant_ values at the 90% confidence level based on a two-sided t-test. We apologize for the mistake and have updated all the captions and discussions accordingly.

In terms of the significance of the nonlinearity in the climate responses (i.e., total vs. WH+EH), the related discussion is adjusted to be the following:

"*The linear summation of WestFF and EastFF results (denoted as "WestFF+EastFF") presented in Fig. 2d shows greater SST responses in the tropical Pacific and tropical Indian Ocean compared with actual FF results (Fig. 2c). However, most of the warming signals over the central and eastern tropical Pacific calculated from WestFF+EastFF are statistically insignificant. Notably, the equatorial West Pacific (160°E–180°) exhibits a significant warming signal in WestFF+EastFF, which is missing in FF results. This disagreement between FF and WestFF+EastFF is likely due to the nonlinear interactions between the impacts of EastFF and WestFF. Similarly, in the extratropical North Pacific, WestFF+EastFF closely resembled the EastFF pattern (i.e., the cooling trend in the western*

*extratropical Pacific; Fig. 2a) while the actual FF results are dominated by WestFF (Fig. 2b&c). The Atlantic response appears to be largely consistent despite greater magnitudes in WestFF+EastFF, with warming from both EastFF and WestFF. Besides the nonlinearity issue, aerosol forcings outside the two focused regions (i.e., aerosols in Africa and the Arabian Peninsula; see Fig. 1d) could also partially contribute to the differences between FF and WestFF+EastFF, particularly by driving cooling over the western tropical Indian Ocean and weakening the wind anomalies (Fig. 2c&d). Additionally, the aerosols outside the focused region could also impose a remote impact on the tropical Pacific region (Huang et al., 2021; Shi et al., 2022), but such impacts are likely to be smaller compared to the nonlinear interactions between EastFF and WestFF impacts, given the small magnitude of the radiative forcing (Fig. 1d).*"

The reviewers also highlighted the recent decline in Chinese aerosol emissions, which isn't captured in the emission dataset used in the CESM1 experiments. You have added some text around line 244 in the tracked changes version of the manuscript. However, while I agree with the reviewers that the emission dataset used in the simulations makes comparison with the real world difficult, I note that you are using ERA5 as 'observations'. I suspect that ERA5 is also using an emission dataset that does not capture the recent decrease in Chinese emissions. Please check what was done with aerosol emissions in ERA5 and update your discussion of the comparison between CESM1 and ERA5 accordingly. In the event that both CESM1 and ERA5 do not capture the recent decline in Chinese emissions, I recommend including a sentence or two about the implications of this for using your analysis to interpret recent observed trends in the Pacific.

**Response**

Thanks for the constructive suggestion.

We checked the technical details of ERA5 (Hersbach et al., 2020), and yes, ERA5 uses aerosol forcings from CMIP5, which is the same as CESM1 experiments applied in this study. However, we note that ERA5 also assimilated many other observational datasets, including shortwave fluxes and sea surface temperature. Therefore, the negative effect of the bias aerosol forcing (especially the unestimated Chinese aerosol decline since the 2010s) on its fidelity may not be as severe as in free-running climate model simulations.

We now updated the related statement in the revised manuscript as follows:

*"One caveat to be noted is that the aerosol forcing scenario (RCP8.5) used in both the Fix_EastFF1920 experiments and ERA5 reanalysis dataset has been shown to overestimate the aerosol emission level in East Asia and miss its observed reduction since the early 2010s despite remaining at a high level (Wang et al., 2021; Xiang et al., 2023). This leads to the overestimation of the EastFF forcing in our model experiments than the real world. On the other hand, similar aerosol forcing biases exist in the ERA5 reanalysis dataset (Hersbach et al., 2020), although the negative impact is mitigated by actually assimilating radiation flux measure and surface temperature. Therefore, although the*

*simulation results and ERA5 appear largely comparable, cautions should be taken in using them to quantitatively interpret the remote impact of Asian aerosol on the North Pacific in the recent couple of decades. In addition, the South Asia emission trend largely follows the assumed emission scenario, which leads to a dipole of aerosol forcings changes within the EH. The forcing dipole might introduce complex circulation responses and lead to different responses over the North Pacific. Simulations with more updated aerosol emission inventory and forcing trends and new observational datasets are necessary to fully explore the realistic climate responses to Asian aerosol forcings."*

I also have an additional question. At L157 you note that your Aleutian Low response doesn't agree with that presented by Smith et al., and attribute this to the use of global aerosol perturbations by Smith et al. (from the CMIP5 DAMIP experiments). I would be curious to see the comparison between the SLP anomalies in your FF simulation and those from Smith et al. Based on the temperature response, I expect this won't change your conclusion. However, it would be interesting to see the comparison between the SLP response in FF, EastFF, WestFF, and WestFF+EastFF.

**Response**

Thanks for the great question!

The SLP changes in FF, EastFF, WestFF, and BMB are already shown in the bottom panels of Fig. 5 and Fig. 6, and we include a combined figure here that only contains SLP in this response for an easier comparison (Fig. R1). The North Pacific SLP trend in response to FF does not match the "aerosol-only" results in Smith et al. (2016) (see Fig. R2). However, it is not surprising because the CMIP5 "aerosol-only" experiment contains both fossil-fuel-burning aerosol (FF) and biomass-burning aerosol (BMB).

Using the same CESM1 single forcing large ensemble experiments, Deser et al. (2020) used the simple linear addition of FF and BMB to represent climate responses to *total* "aerosol-only" forcings. Similar to the CMIP5 "aerosol-only" results (Fig. 2b), FF+BMB shows an increase in SLP over the North Pacific. This supports our statement in the manuscript regarding the competing role of FF and BMB.

However, different from the large-scale SLP increase over the North Pacific in the CMIP5 results, CESM1's positive SLP changes in response to FF+BMB cover smaller regions and with smaller magnitudes. This is likely due to the nonlinear interactions between FF and BMB impacts (similar to the discussed nonlinear interactions when adding EastFF and WestFF responses; see the disagreement between Fig. R1 c, and d).

In summary, the Editor raised a very interesting (and important) question that is worth more discussion. However, since we intend to focus our discussions on the comparisons between EastFF and WestFF (which are based on our own regional forcing simulations), we

are keen on keeping the detailed discussions here rather than including them in the manuscript. Instead, We include Fig. R1 as the new Fig.S1 in the supplementary document, and updated our SLP-related statements in the revised manuscript as follows:

*"Smith et al. (2016) argued that the AA forcing originating from the East Asia region induces large-scale warming in the North Pacific and leads to a weakening of the Aleutian Low, while our results here, however, do not support such an argument. The reason is that only the total aerosol-only experiments from CMIP5 were analyzed in the previous study, which indeed yields consistent responses when compared to the FF+BMB results in the CESM1 single forcing large ensemble experiments (Deser et al.(2020); also shown in Fig. S1 in the supplement despite a weaker magnitude). Therefore, we suggest the nonlinearity when combining regional (e.g., EH and WH) or sectoral (e.g., FF and BMB) aerosol responses makes it very challenging to clearly distinguish the climate impact due to aerosol forcings from particular sources or regions simply based on those total "aerosol-only" experiments as in CMIP5/6's DAMIPs, hence justifying the importance of running more nuanced regional aerosol perturbation experiments as designed and conducted here in our study."*

[Figure]

**Fig. R1**

*Decadal changes in sea level pressure (shading; Pa per decade) during 1980–2020 calculated in response to (a) EastFF, (b) WestFF, (c) FF, and (e) BMB. (d) shows the decadal changes in sea level pressure calculated from the linear addition of EastFF and WestFF. (f) is similar to (d) but shows the results from the linear addition of FF and BMB. Stippled regions indicate significant values at the 90% confidence level based on a two-sided t-test.*

[Figure]

**Fig. R2**

*Decadal trend of sea level pressure (hPa per decade) from (a) observation and (b) CMIP5 model simulations. Figure borrowed from Figure 3 in Smith et al. (2016).*

Please also address a few remaining typos etc. All line numbers refer to the marked up version.

L15: 'Based on CESM1' -> 'with CESM1'

— We changed words as suggested.

L59: Revert to your original text here, which was correct. There are a few places throughout the manuscript where citations are incorrectly formatted. Please check for more (e.g. L155, L163, ...)

— Fixed.

L96: additionally -> Additionally

— Fixed.

L174: Depending on the mechanism, the Pacific could also be considered to be downstream. I suggest simply deleting this comment.

—Thanks for the suggestion. We deleted this comment as suggested.

**References**

Hersbach, H., Bell, B., Berrisford, P., Hirahara, S., Horányi, A., Muñoz-Sabater, J., Nicolas, J., Peubey, C., Radu, R., Schepers, D., Simmons, A., Soci, C., Abdalla, S., Abellan, X., Balsamo, G., Bechtold, P., Biavati, G., Bidlot, J., Bonavita, M., De Chiara, G., Dahlgren, P., Dee, D., Diamantakis, M., Dragani, R., Flemming, J., Forbes, R., Fuentes, M., Geer, A., Haimberger, L., Healy, S., Hogan, R. J., Hólm, E., Janisková, M., Keeley, S., Laloyaux, P., Lopez, P., Lupu, C., Radnoti, G., de Rosnay, P., Rozum, I., Vamborg, F., Villaume, S., and Jean-Noël Thépaut: The ERA5 global reanalysis, Q. J. R. Meteorol. Soc., 146, 1999–2049, 2020.

Deser, C., Phillips, A. S., Simpson, I. R., Rosenbloom, N., Coleman, D., Lehner, F., Pendergrass, A. G., DiNezio, P., and Stevenson, S.: Isolating the evolving contributions of anthropogenic aerosols and greenhouse gases: A new CESM1 large ensemble community resource, J. Clim., 33, 7835–7858, 2020.

Smith, D. M., Booth, B. B. B., Dunstone, N. J., Eade, R., Hermanson, L., Jones, G. S., Scaife, A. A., Sheen, K. L., and Thompson, V.: Role of volcanic and anthropogenic aerosols in the recent global surface warming slowdown, Nat. Clim. Chang., 6, 936–940, 2016.